# Structural insights into the antibacterial function of the *Pseudomonas putida* effector Tke5

Carmen Velázquez [ID] [1,2,5], Maialen Zabala-Zearreta [ID] [1,5], Carmen Paredes[3], Cristina Civantos[3], Jon Altuna-Alvarez [ID] [1], Patricia Bernal [ID] [3✉] & David Albesa-Jové [ID] [1,2,4✉]

## Abstract

*Pseudomonas putida* **is a plant-beneficial rhizobacterium that encodes multiple type-VI secretion systems (T6SS) to outcompete phytopathogens in the rhizosphere. Among its antibacterial effectors, Tke5 (a member of the BTH_I2691 protein family) is a potent pore-forming toxin that disrupts ion homeostasis without causing considerable membrane damage. Tke5 harbours an N-terminal MIX domain, which is required for T6SS-dependent secretion in other systems. Many MIX domain-containing effectors require T6SS adaptor proteins (Tap) for secretion, but their molecular mechanisms of adaptor-effector binding remain elusive. Here, we report the 2.8 Å cryo-EM structure of the Tap3-Tke5 complex of** *P. putida* **strain KT2440, providing structural and functional insights into how effector Tke5 is recruited by its cognate adaptor protein Tap3. Functional dissection shows that the α-helical region of Tke5 is sufficient to kill intoxicated bacteria, while its β-rich region likely contributes to target membrane specificity. These findings delineate a mechanism of BTH_I2691 proteins for Tap recruitment and toxin activity, contributing to our understanding of a widespread yet understudied toxin family.**

**Keywords** Bacterial Toxins; Bacterial Structural Biology; Pore-forming Toxins; Type VI Secretion System; BTH_I2691 (VasX) Family
**Subject Categories** Membranes & Trafficking; Microbiology, Virology & Host Pathogen Interaction; Structural Biology

## Introduction

*Pseudomonas putida* KT2440 is a plant growth-promoting rhizobacterium of major interest in agricultural biotechnology due to its stimulatory and protective effects on plants. This strain exhibits several beneficial traits, including the production of indole compounds and siderophores, phosphate solubilisation, and 1-aminocyclopropane-1-carboxylate (ACC) deaminase activity. These features have been shown to enhance germination rates, increase root and shoot length, improve fresh and dry biomass, and trigger induced systemic resistance in plants (Costa-Gutierrez et al, 2022).

*P. putida* KT2440 harbours three distinct Type VI secretion system (T6SS) gene clusters, designated K1-, K2-, and K3-T6SS (Bernal et al, 2017). The T6SS is a sophisticated contractile nanomachine employed by many Gram-negative bacteria to deliver toxic effector proteins into neighbouring cells, thereby providing a competitive advantage in complex microbial communities (Mougous et al, 2006; Russell et al, 2011; Vázquez-Arias et al, 2025). In *P. putida* KT2440, the K1-T6SS has been shown to effectively eliminate a broad range of bacterial competitors, including recalcitrant phytopathogens, thereby playing a key role in preventing plant infections and enhancing the bacterium's biocontrol activity (Bernal et al, 2017, 2021).

Among the effectors encoded within the K3-T6SS cluster of *P. putida* KT2440, a recent study identified a Type six KT2440 effector 5 (Tke5) as a potent antibacterial toxin effective against ten of the most resilient plant pathogens (Velázquez et al, 2025). Notably, this work also elucidated Tke5's molecular mechanism of action, showing that it functions as a novel pore-forming toxin (PFT) that kills target cells through selective ion transport. Instead of causing extensive membrane disruption, Tke5 induces membrane depolarisation by perturbing ion homeostasis, ultimately leading to bacterial cell death while preserving overall membrane integrity (Velázquez et al, 2025). This work represented the first biophysical dissection for a member of the T6SS effector BTH_I2691 protein family (InterPro accession NF041559), also commonly referred to as the VasX family. Members of this family are large, multi-domain pore-forming toxins harbouring an N-terminal marker for type six secretion system effectors (MIX) motif that was shown to be necessary for T6SS-dependent secretion (Salomon et al, 2014; Fridman et al, 2022). Those family members that have been experimentally studied to date illustrate a remarkable evolutionary diversification in their membrane-targeting strategies. These include the archetypal *Vibrio cholerae* effector VasX, a toxin capable of killing both eukaryotic cells, such as the amoeba *Dictyostelium discoideum*, and prokaryotic competitors by engaging membranes via direct lipid-binding (Miyata et al, 2011, 2013); the structural prototype *Pseudomonas aeruginosa* Ptx2 (Colautti et al, 2025); and the highly divergent *Bacteroides fragilis* effector Bte2, a

[1]Instituto Biofisika (CSIC, UPV/EHU), Fundación Biofísica Bizkaia/Biofisika Bizkaia Fundazioa (FBB), Leioa 48940, Spain. [2]Departamento de Bioquímica y Biología Molecular, University of the Basque Country, Leioa 48940, Spain. [3]Departamento de Microbiología, Facultad de Biología, Universidad de Sevilla, Seville 41012, Spain. [4]Ikerbasque, Basque Foundation for Science, Bilbao 48013, Spain. [5]These authors contributed equally: Carmen Velázquez, Maialen Zabala-Zearreta. ✉E-mail: pbernal@us.es; david.albesa@ehu.eus

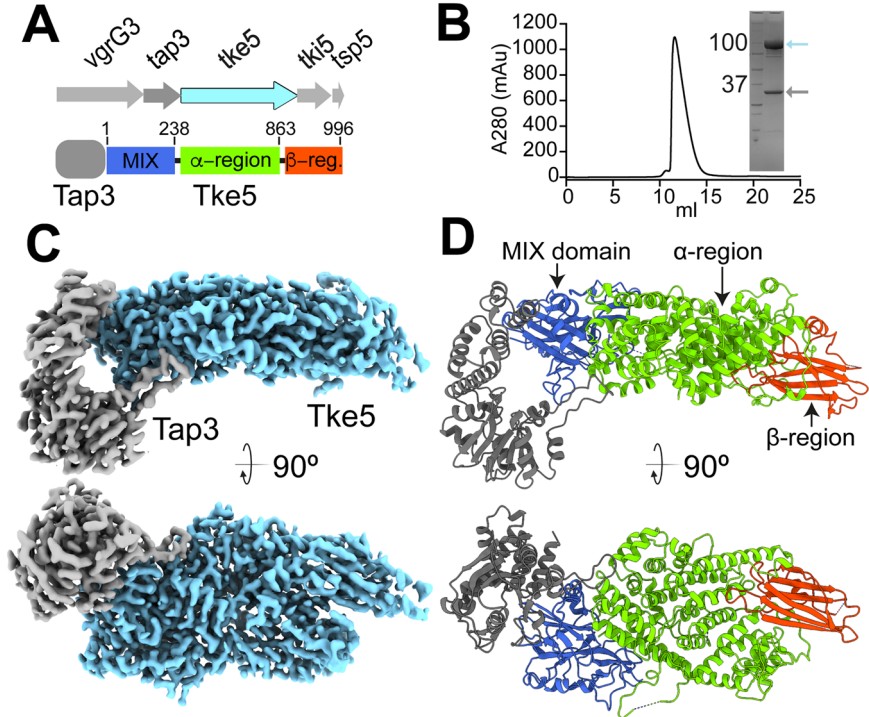

**Figure 1. Genetic context, purification, and cryo-EM structure of the Tap3–Tke5 complex.**

(A) Top, schematic genetic organisation of the *vgrG3*, *tap3*, *tke5*, *tki5*, and *tsp5* genes. *tap3* and *tke5* are co-expressed from the pCOLADuet-1 plasmid, while *tki5* is expressed from the pSEVA624C plasmid to neutralise potential Tke5-induced toxicity. The arrows indicate the direction of transcription. Bottom, linear representation of Tap3–Tke5 complex showing the domain organisation of Tke5 and its corresponding residue boundaries. (B) Gel filtration chromatogram of the purified Tap3–Tke5 complex. The complex eluted as a single, monodisperse peak, indicating a stable and homogeneous sample. The x-axis represents the elution volume (mL), and the y-axis represents the absorbance at 280 nm (mAU). The inset shows an SDS-PAGE analysis of the purified Tap3–Tke5 complex. The gel shows two distinct bands, corresponding to Tke5 and Tap3, confirming the presence of both proteins in the purified complex. This result is consistent with the mass spectrometry data (Dataset EV1). Source data for this figure are available online. (C) Cryo-EM density of Tap3–Tke5 complex displayed perpendicular to the longest axis of the complex and rotated 90° clockwise (map post-processed with DeepEMhancer). (D) Cartoon representation of the Tap3–Tke5 atomic structure displayed perpendicular to the longest axis of the complex and rotated 90° clockwise. Tap3 is shown in grey, and Tke5 is coloured in blue (MIX domain), green (α-helical region), and red (β-rich region). Source data are available online for this figure.

gut-adapted specialist that has uniquely co-opted the Rnf electron transport complex in target cells as a receptor to mediate intoxication (Ratner et al, 2025).

The *tke5* gene is found near the 3' end of the K3-T6SS cluster (Fig. 1A), where genes coding for a Valine-glycine repeat protein G (VgrG3), a Type VI adaptor protein (Tap3), Tke5, its cognate Type six KT2440 immunity protein (Tki5), and a Type six paar (Tsp5) are also found (Bernal et al, 2017; Velázquez et al, 2025). VgrG proteins form a trimeric complex at the distal end of the Haemolysin coregulated protein (Hcp) tube and are essential for the assembly and function of the T6SS (Hachani et al, 2014). The last gene in the cluster, *tsp5*, is predicted to code for a PAAR (proline-alanine-alanine-arginine repeat) protein. PAAR proteins assemble cone-shaped structures that sharpen and diversify the VgrG spike complex (Shneider et al, 2013).

Antiprokaryotic effector proteins delivered by the T6SS are often paired with cognate immunity proteins that protect the producing cell from self-intoxication, as well as interspecies (Basler et al, 2013; Hood et al, 2010) and intraspecies (George et al, 2024; Unterweger et al, 2012) competition. In this regard, previous work has demonstrated that Tki5 expression protects from Tke5-induced toxicity (Velázquez et al, 2025).

Tap3 is a protein belonging to the DUF4123 family, which has recently been identified as a group of T6SS adaptor proteins that play a crucial role in enabling the recognition and export of evolutionarily unrelated effectors (Colautti et al, 2024; Unterweger et al, 2015). DUF4123 proteins are often found downstream of *vgrG* or upstream of putative effector-encoding genes in predicted T6SS gene clusters. The presence of *vgrG3*, *tap3*, *tke5*, *tki5*, and *tsp5* genes within the K3-T6SS cluster underscores the complex and coordinated nature of the T6SS machinery, highlighting how structural components, adaptors, effectors, and immunity proteins work together to enable the system's function.

Previous work on a Tap from *Pseudomonas aeruginosa* (Tap6) demonstrated that it directly binds its cognate effector, Ptx2. Experiments confirmed its essential role in Ptx2 export and suggested that Tap6's variable C-terminal lobe mediates effector recognition (Colautti et al, 2024). Nonetheless, despite the importance of T6SS adaptor proteins, the molecular mechanisms they use to bind to their cognate effectors have not been experimentally validated. To bridge this knowledge gap, we have determined the 2.8 Å cryo-EM structure of Tke5 in complex with its cognate adaptor Tap3. Our structure uncovers how Tap3 recognises the MIX domain of Tke5. Furthermore, we define the

domain architecture of Tke5 and demonstrate that only an α-helical region located towards its C-terminal end is sufficient to kill intoxicated bacteria. In contrast, a β-rich region at the C-terminus is likely to enhance target membrane specificity. Together, our results provide a comprehensive framework for understanding how similar MIX-containing BTH_I2691 effectors might be recognised by their cognate Tap proteins, and we dissect their structural organisation and mechanism of action.

# Results

## A 2.8 Å cryo-EM structure of the Tap3–Tke5 complex provides insight into Tke5's molecular mechanisms for T6SS-loading and toxicity

All characterised DUF4123 proteins function as adaptor proteins, providing a direct and essential physical connection between VgrG proteins and diverse families of cognate effectors that lack PAAR or "PAAR-like" N-terminal domains (Pei et al, 2020; Burkinshaw et al, 2018; Liang et al, 2015; Miyata et al, 2013; Unterweger et al, 2015; Colautti et al, 2024). Given the genomic context of the *tap3* and *tke5* genes (Fig. 1A) and the function of other Tap proteins, we expected a direct interaction between Tap3 and Tke5 proteins. To demonstrate this, we have solved the 2.8 Å-resolution structure of the Tap3-Tke5 complex using cryo-electron microscopy (cryo-EM) (Fig. 1; Movie EV1; Appendix Figs. S1–S3; Table 1). To this end, a 6xHis tag was encoded at the 5' end of *tap3* (*6xhis-tap3*). This construct was inserted at the Multiple Cloning Site 1 (MCS-1) of a pCOLADuet™-1 plasmid. Then, the unmodified *tke5* gene was inserted at the MCS-2 of pCOLADuet™-1. The resulting plasmid pCOLADuet-1•*6xhis-tap3-tke5* was transformed into *Escherichia coli* BL21(DE3) for heterologous co-expression of Tap3 and Tke5 (see "Methods" for details). Following co-expression, we co-purify the Tap3–Tke5 complex to homogeneity in milligram quantities (Fig. 1B). The resulting cryo-EM density map shows high quality, permitting de novo atomic model building of the complex except for residues 1–12 (N-terminus) and 303–307 (C-terminus) of Tap3, and 1–16 (N-terminus), 267–273, 730–757, and 886–909 of Tke5.

Analysis of the Tap3–Tke5 complex reveals that Tap3 comprises two domains that assemble into a horseshoe-like fold (Fig. 2A), representing a novel fold without significant representatives in the Protein Databank (based on a Foldseek search (Kim et al, 2025)). The Tap3 N-terminal domain is an α/β-domain assembled by a central anti-parallel β-sheet containing six β-strands (shown in red) surrounded by a β-hairpin and six helices (three helices on each face of the β-sheet). The C-terminal domain is an α-helical bundle assembled by five helices (see Appendix Fig. S4 for the secondary structure arrangement).

Tke5 is organised into two domains: an N-terminal α/β domain containing the MIX motif (residues [17–238]; shown in blue in Fig. 2B) and an α + β domain (residues [239–996]; shown in green and red). See Appendix Figs. S5 and S6 for its secondary structure arrangement. The MIX motif-containing domain (hereafter named the MIX domain) mediates most of the binding with Tap3.

Remarkably, Tap3 includes a large loop (residues 31–50, named thereafter Tap3-Loop[31–50]) that protrudes outwards and accounts for ~54% of the Tap3 residues that bind to Tke5 (Fig. 3A,B).

**Table 1. Cryo-EM data collection, data processing, and model statistics.**

| Data collection | EMD 53820, PDB 9R8G, EMPIAR 13034 |
|---|---|
| Microscope | ThermoFisher Titan Krios |
| Magnification | 130,000× |
| Voltage (kV) | 300 |
| Camera | Gatan K3 |
| Pixel size (Å/pixel) | 0.6462 |
| Total electron dose (e⁻/Å²) | 48.8 |
| Exposure time (s) | 1.3 |
| Defocus range (µm) | −0.8 to −2.5 |
| Number of images | 19,112 |
| **Processing** | |
| Number of selected micrographs | 18,617 |
| Number of initial particles | 3,054,061 |
| Number of final particles | 381,732 |
| Final resolution (Å) | 2.8 |
| Symmetry | C1 |
| Map sharpening B factor (Å²) | 100.8 |
| **Orientation diagnostics** | |
| SCF* | 0.827 |
| cFAR | 0.57 |
| **Structure composition/validation** | |
| No. of chains | 2 |
| Non-hydrogen atoms | 8811 |
| Protein residues | 1211 |
| Bond RMSD lengths (Å) (#>4 σ) | 0.004 (0) |
| Bond RMSD angles (°) (#>4 σ) | 0.452 (3) |
| Molprobity score | 1.15 |
| Clashscore | 3.42 |
| Rotamers outliers (%) | 0.12 |
| Ramachandran favoured (%) | 97.92 |
| Ramachandran allowed (%) | 2.08 |
| Ramachandran outliers (%) | 0 |
| CC (mask) | 0.86 |
| CC (box) | 0.87 |
| CC (peaks) | 0.78 |
| CC (volume) | 0.85 |

Besides Tap3-Loop[31–50], two more regions in Tap3 bind to Tke5, one of which is also located within the N-terminal domain (residues R75, E77, F78; Fig. 3B); and a second region within the C-terminal α-helical bundle (residues N251, L284, E286, S287, P288, Q289, A290, R293; Fig. 3C). In summary, the N- and C-terminal domains of Tap3 bind to the MIX domain of Tke5, except Tap3-Loop[31–50] that extends its binding beyond the Tke5's MIX domain and into Tke5's α + β-domain. All hydrogen bonds between Tap3 and Tke5 are represented in Appendix Fig. S7 and Appendix Table S1.

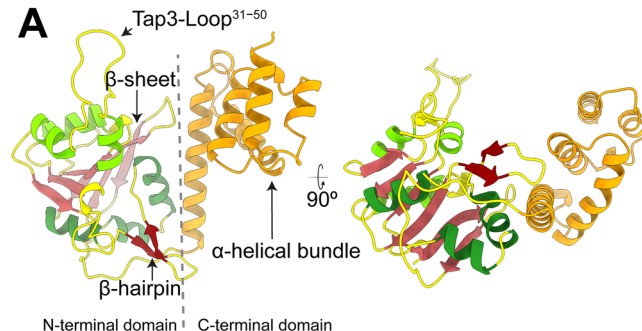

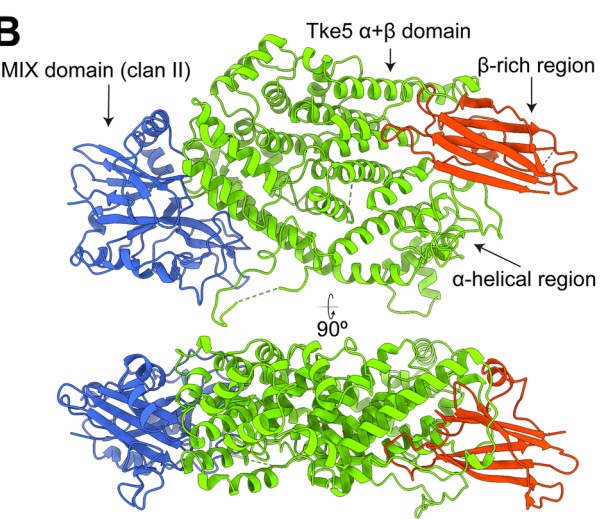

**Figure 2. Overall domain architecture of Tap3 and Tke5.**

(A) Two 90° rotated views of Tap3 shown as a cartoon representation, highlighting its N-terminal and C-terminal domains. The N-terminal domain (residues 1–198) adopts an α/β fold, featuring a central anti-parallel β-sheet (shown in red) flanked by a β-hairpin (shown in dark red) and six α-helices (three on each side of the β-sheet). The C-terminal domain (residues 199–302) forms an α-helical bundle comprising five helices (shown in orange). (B) Two views rotated 90° of Tke5 are shown as cartoon representations, highlighting its N-terminal and C-terminal domains. The N-terminal α/β domain contains a MIX fold (residues [17–238] shown in blue). The C-terminal α + β domain (residues [239–996]) is organised in an α-region (residues [239–863] shown in green) and a β-rich region (residues [864–996] shown in red). The β-rich region presents an immunoglobulin-like sandwich fold.

## The MIX fold is a conserved structural scaffold for effector specificity and adaptor recruitment

The MIX motif is commonly found in antibacterial T6SS effectors, and sequence clustering analyses have identified up to five distinct clans (MIX I to MIX V) (Salomon et al, 2014). This MIX motif was defined by a conserved central sequence, hRxGhhYhh (where *h* denotes a hydrophobic residue), flanked by less conserved segments at the N-terminus (shhPhR) and the C-terminus (hhF/YSxxxWS/T) (Salomon et al, 2014). The cryo-EM structure of Tke5 reveals that the MIX fold has no close structural homologues in the Protein Data Bank (as determined by Foldseek (Kim et al, 2025) analysis). This fold features a distinctive pyramid-like architecture composed of two central β-sheets connected by variable-length loops and six α-helices (Fig. 4A,B; Appendix Fig. S8).

The specific sequence of the MIX motif in Tke5 comprises residues [20]PILPVR[25], [64]LRPGYVYVF[72], and [132]IGYSPHLWT[140]. R25 and R65 residues are forming salt bridges with the α + β-domain; Y68, L138, and T140 are involved in hydrogen bonds with the α + β-domain; and P66, G67, Y68, Y134, P136, H137, L138, W139, and T140 are also in contact with the α + β-domain (Fig. 4B). Thus, the structure reveals that most of the residues in the MIX motif localise at the interface between MIX and α + β-domains, which suggests that MIX domains might provide specificity to their toxic domains found in MIX-containing T6SS effectors, as well for recognising their corresponding cognate Tap proteins. This latter function has been recently demonstrated for two *Pseudomonas aeruginosa* adaptor proteins, Tap6 and Tap14, which specifically tether unrelated T6SS effectors to their respective VgrG partners (Colautti et al, 2024). All hydrogen bonds between the MIX domain and the α + β domain are represented in Appendix Fig. S9 and Appendix Table S2.

## The α-helical region of Tke5 is essential for toxicity and immunity-mediated neutralisation

The toxic activity encoded within Tke5 is responsible for bacterial killing, membrane depolarisation, and pore formation (Velázquez et al, 2025). Based on Tke5's domain architecture, the pore-forming activity is likely mediated by its C-terminal α + β domain. This domain can be subdivided into two distinct regions: an N-terminal α-helical region (residues 239–863 shown in grey and green; Fig. 5A) and a C-terminal β-rich region (residues 864–996 shown in red; Fig. 5A,B). The α-helical region is assembled by 34 helices (~62% of residues). These helices are closely packed, forming a compact structure. Prediction of transmembrane helices based on primary sequence suggests that five segments within the α-helical region might have a propensity for membrane insertion (predicted by TMHMM 2.0 (Krogh et al, 2001); Appendix Fig. S10). These five segments are contained within residues 608–813 and include all or part of 11 helices. Six of these helices are buried in the core of the α-helical region, while the remaining five are located on the surface (residues 608–813 shown in green in Fig. 5A; Appendix Table S3). Given that this structure is most likely representing a pre-pore state of Tke5 found in solution, one would expect major conformational changes upon membrane insertion, which might drive some of these helices to insert into the hydrophobic core of the target membrane, resulting in the assembly of a pore state.

The C-terminal β-rich region presents an immunoglobulin-like sandwich fold. The sandwich is assembled by two anti-parallel β-sheets, one with three strands and the other with four. One face of the sandwich is packed against the only helix in this domain (residues 864–996 shown in red; Fig. 5A,B). Tke5 homologues are found across the Pseudomonadota (previously known as Proteobacteria) phylum, where they are predominantly annotated as T6SS effector BTH_I2691 family proteins. Interestingly, some family members do not contain the C-terminal β-rich region, which led us to hypothesise that the α-helical region might be sufficient for toxicity.

To test our hypothesis, we measure the toxicity of Tke5 full-length and deletion constructs when expressed from plasmid pS238D•M in *P. putida* KT2440 (Fig. 5C,D). We engineer the following constructs: Tke5 wild-type (MIX-α-β[1–996]) and a deletion construct lacking the C-terminal β-rich region (MIX-α[1–863]).

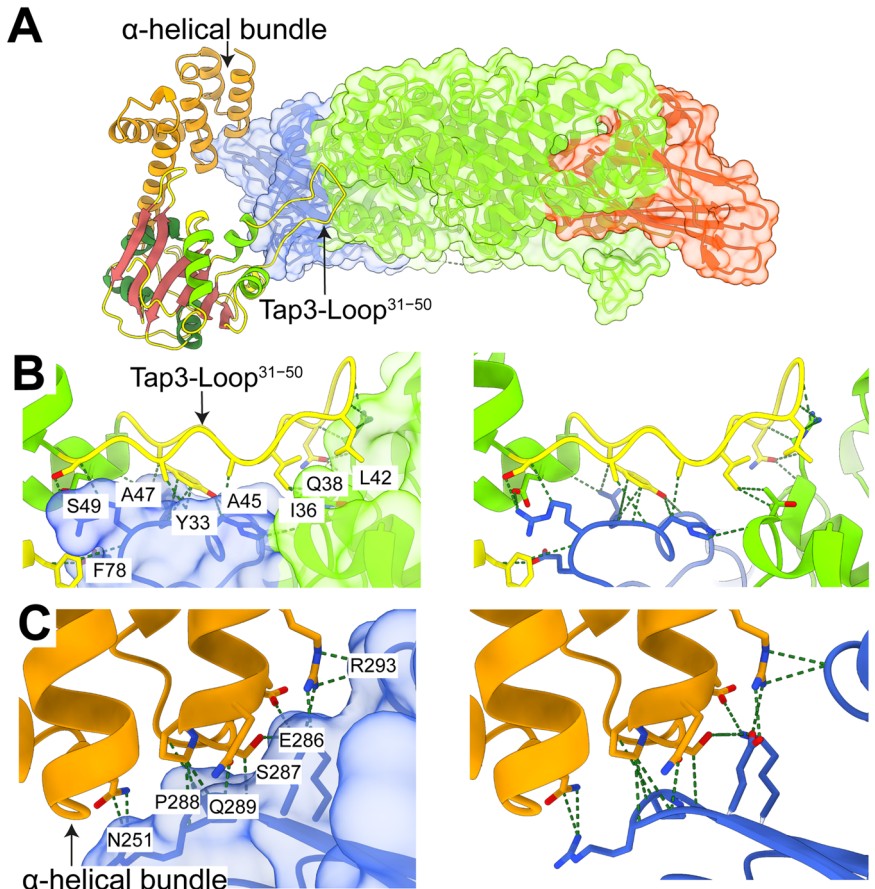

**Figure 3. Structural insights into Tap3–Tke5 interactions.**

(A) Cartoon representation of the Tap3–Tke5 atomic structure. The solvent-excluded molecular surface of Tke5 is also shown. The colouring scheme is the same as for Fig. 2: the Tke5's MIX fold is shown in blue, the α-region is shown in green, and the β-rich region is shown in red. Tap3's N-terminal domain is colour-coded by secondary structure elements, and the C-terminal α-helical domain is shown in orange. (B) Close-up view of the prominent loop (Tap3-Loop$^{31-50}$) extending from the N-terminal domain of Tap3 and interacting with Tke5. This loop protrudes outward and accounts for 54% of the Tap3 residues contacting Tke5. The solvent-excluded molecular surface of Tke5 is shown on the left panel and is hidden on the right panel. (C) Close-up view showing the interactions between Tap3's C-terminal α-helical domain and Tke5's MIX fold. The solvent-excluded molecular surface of Tke5 is shown on the left panel and is hidden on the right panel.

Furthermore, we engineer Tke5 constructs containing an N-terminal PelB leader sequence (PelB·MIX-α-β$^{1-996}$) and deletion constructs PelB·MIX$^{1-238}$, PelB·α-β$^{239-996}$, PelB·α$^{239-863}$, PelB·β$^{864-996}$.

Previously established, the expression of Tke5 is toxic to *P. putida* KT2440, an effect that is neutralised by the co-expression of its cognate immunity protein, Tki5. The toxin's potency is significantly enhanced by the addition of a PelB leader sequence, which targets the protein for secretion into the periplasm (Velázquez et al, 2025). This increased toxicity is the expected outcome, as it aligns with previous in vitro electrophysiological data showing that Tke5 senses the directionality of the membrane potential and is expected to preferentially insert into the inner membrane from the periplasmic side (Velázquez et al, 2025).

To pinpoint the toxic component of Tke5, we have systematically analysed the series of deletion constructs generated. Our findings show that the C-terminal β-rich region is not essential for toxicity, as its deletion (PelB·α$^{239-863}$ or MIX-α$^{1-863}$) does not eliminate the toxic effect (Fig. 5C,D). Furthermore, constructs lacking the N-terminal MIX domain (PelB·α-β$^{239-996}$ or PelB·α$^{239-863}$) remain as toxic as the full-length protein PelB·MIX-α-β$^{1-996}$, while

the MIX domain (PelB·MIX$^{1-238}$) or the C-terminal β-rich region (PelB·β$^{864-996}$) alone are non-toxic (Fig. 5C). Crucially, the minimal construct containing only the α-helical region (PelB·α$^{239-863}$) is sufficient to induce cell death, demonstrating that this domain is the core toxic module of the effector (Fig. 5C).

To validate that this observed toxicity is the result of Tke5's specific activity, we have co-expressed the toxic constructs with the cognate immunity protein (Fig. 5D). The presence of Tki5 fully protected the cells from the effects of full-length Tke5 (MIX-α-β$^{1-996}$, PelB·MIX-α-β$^{1-996}$), and deletion constructs containing the α-helical region (MIX-α$^{1-863}$, PelB·α$^{239-863}$). Taken together, these results confirm that the immunity protein acts by directly neutralising the α-helical region.

In summary, while previous work identified Tke5 as a bactericidal toxin that causes cell depolarisation, our findings now establish that this activity is mediated entirely by its α-helical region. This region alone is sufficient to kill intoxicated bacteria, and the ability of Tki5 to neutralise its effect confirms that the immunity protein directly counteracts the effector's toxic activity rather than interfering with other domains.

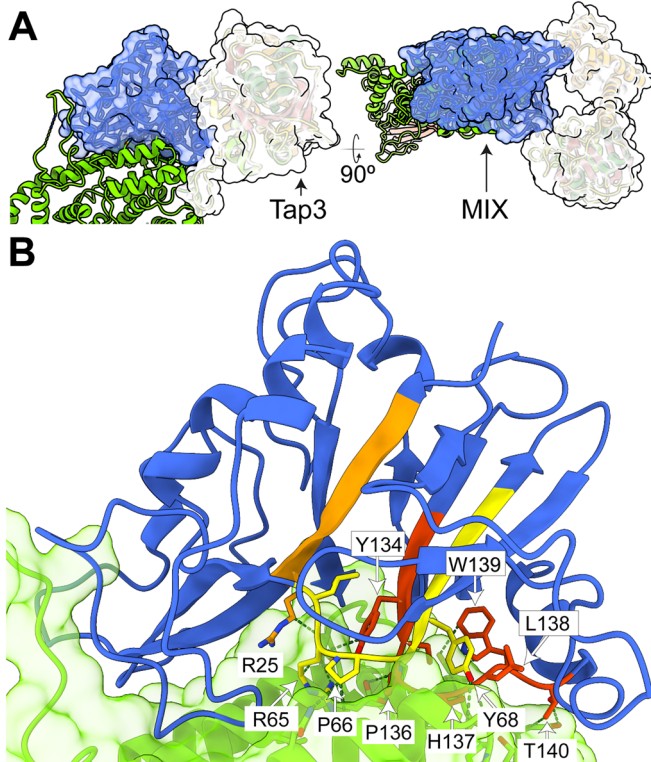

**Figure 4. The MIX motif and its role in Tke5 domain specificity.**

(A) Two 90° anti-clockwise rotated views of the Tap3–Tke5 atomic structure. The solvent-excluded molecular surface of the MIX domain and Tap3 are also shown. The colouring scheme is the same as for Figs. 2 and 3. (B) Close-up view of the MIX–α + β domain interface, highlighting the MIX motif with residues 20–25 (PILPVR) in orange, 64–72 (LRPGYVYVF) in yellow, and 132–140 (IGYSPHLWT) in red. Key interfacial contacts are indicated, including salt bridges formed by R25 and R65, hydrogen bonds involving Y68, L138, and T140, and additional contacts contributed by P66, G67, Y68, Y134, P136, H137, L138, W139, and T140.

## The C-terminal β-rich region of Tke5 binds to model membranes in vitro

Following the experimental demonstration that the α-helical region encodes Tke5' toxicity, we turn our attention to the C-terminal β-rich region (hereafter referred to as the Receptor Binding Domain (RBD)). The predicted position of the cryo-EM Tke5 structure on a model Gram-negative inner membrane by the PPM 3.0 server (Protein Property Prediction and Modelling Server (Lomize et al, 2022); see "Methods" for details) suggests that Tke5 can bind to the membrane via the RBD domain (Fig. 6A). In particular, this binding is predicted to occur through the most distal end of the RBD, including a disordered loop (residues 886–909) not visible in the cryo-EM map that we modelled using AlphaFold_unmasked (Mirabello et al, 2024). The predicted interaction occurs with one of the leaflets of the membrane, suggesting an initial, perhaps low-affinity, interaction with the lipid bilayer. This membrane-sensing role of the RBD could be relevant for guiding Tke5 to its target inner membrane and potentially orienting it for subsequent insertion.

To experimentally test this hypothesis, we used a fluorescence co-localisation assay to investigate the RBD's ability to bind model membranes. We prepared rhodamine-labelled Giant Unilamellar Vesicles (GUVs) from two distinct lipid compositions: a complex *E. coli* polar lipid extract (PE:PG:CL at 67.0%, 23.2%, and 9.8%, respectively) and simple, pure 1,2-dioleoyl-sn-glycero-3-phosphocholine (DOPC; see "Methods" for details). Upon incubation, fluorescence microscopy revealed that while a GFP-only control remained homogeneously distributed, the signal from an engineered GFP–RBD fusion protein specifically co-localised with the rhodamine signal of the GUVs, indicating a direct interaction with the membrane (Fig. 6B).

These results provide direct experimental validation for the predicted membrane-binding capacity of the Tke5 C-terminal domain. Furthermore, because binding occurred with GUVs composed of both complex *E. coli* lipids and simple DOPC, our findings support the RBD's role as a membrane-tethering module that does not require specific lipid head groups to function.

## Discussion

### Structural insights into Tap-effector recognition and the BTH_I2691 protein family

Our 2.8 Å cryo-EM structure of the Tap3-Tke5 complex from *Pseudomonas putida* KT2440 provides the first atomic-level insight into how DUF4123 adaptor proteins recognise their cognate effectors. This high-resolution structure reveals novel protein folds for both Tap3 and Tke5. Tap3 adopts a unique horseshoe-like bilobed architecture, while Tke5 is organised into an N-terminal α/β domain containing the MIX motif (Clan II) and a C-terminal α + β domain. Crucially, the Tke5 MIX domain mediates the majority of the binding with Tap3. A remarkable feature of this interaction is the prominent Tap3-Loop[31–50], which protrudes from Tap3's N-terminal domain and accounts for approximately 54% of the Tap3 residues involved in binding to Tke5. This loop extends its interaction beyond Tke5's canonical MIX domain into its α + β-domain, indicating a broad and intimate interface.

Previous research had demonstrated a direct interaction between another Tap protein (Tap6) and its cognate MIX-containing BTH_I2691 effector (Ptx2). The work combined AlphaFold 3 (AF3) predictions with extensive experimental validation (Colautti et al, 2024). While AF3 confidently predicted Tap6's bilobed architecture and its N-terminal lobe interacting with a C-terminal helix-turn-helix (HTH) motif found in its cognate VgrG6, it was notably unable to confidently predict the structure of Ptx2 or the complex formed between Ptx2 and Tap6 due to Ptx2 having few close sequence or structural homologues (Colautti et al, 2024). Despite this, the Tap6-Ptx2 binding mechanism was strongly supported by experimental evidence, including bacterial competition assays that demonstrated Tap6's essential role in Ptx2 export, in vitro size exclusion chromatography showing stable complex formation, and in vivo co-immunoprecipitation, which confirmed that Tap6 mediates the interaction between Ptx2 and VgrG6. Based on the general bilobed architecture of DUF4123 proteins, it was proposed that Tap6's variable C-terminal lobe is responsible for recognising and binding Ptx2 (Colautti et al, 2024).

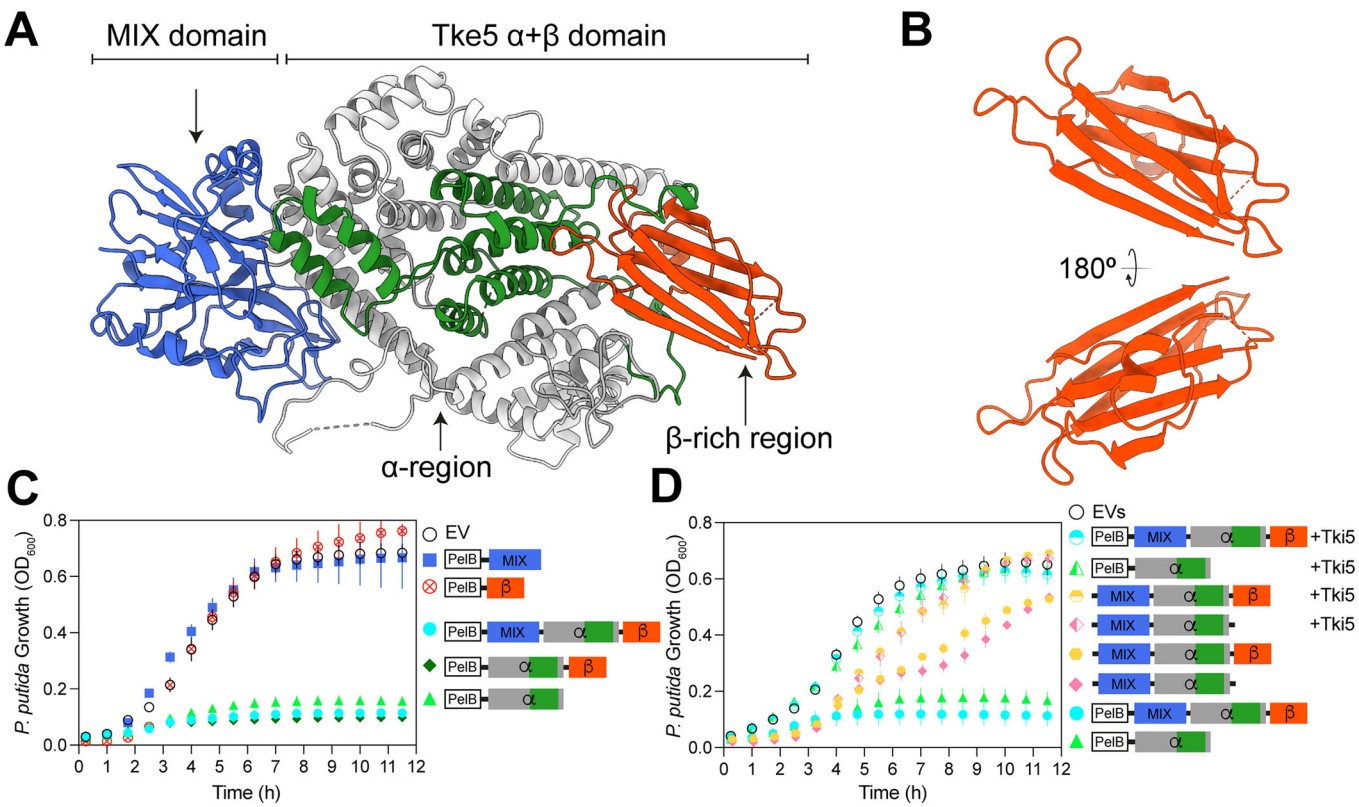

**Figure 5.  The α-helical region of Tke5 mediates toxicity and is neutralised by its cognate immunity protein Tki5.**

(A) Domain architecture of Tke5 showing the N-terminal MIX domain (residues 17–238, blue), α-helical pore-forming region (residues 239–863, grey and green), and C-terminal β-rich region (residues 864–996, green). Predicted transmembrane helices (green) are mapped onto the α-helical region, which corresponds to residues 622-644 (FLGLLTSGGGGGLNAGMLWFNIL), 663-685 (LGFASSVFGVIGAAAATLVSV-RA), 695-717 (ISTAPGMAFGNGIINFLTSNLFA), 744-766 (TALGSILVL) and 781–803 (FAGWTAAGIALIGATLIGGGLFL). (B) Two 180° rotated views of the β-rich region that presents an immunoglobulin-like sandwich fold located at the C-terminal of Tke5. (C) Toxicity assays of PelB-containing Tke5 and truncation constructs in *P. putida*. Full-length Tke5 (PelB·MIX-α-β$^{1-996}$), the α + β domain (PelB·α$^{239-996}$), and the α-helical region alone (PelB·α$^{239-863}$) show comparable toxicity. The MIX domain (PelB·MIX$^{1-238}$) and β-rich region (PelB·β$^{864-996}$) are non-toxic, with bacterial growth comparable to that of bacteria transformed with the empty vector (EV). (D) Neutralisation of Tke5 toxicity by its cognate immunity protein Tki5. Co-expression of Tki5 rescues cell growth inhibited by PelB·MIX-α-β$^{1-996}$, PelB·α$^{239-863}$, MIX-α-β$^{1-996}$ and MIX-α$^{1-863}$, confirming the specificity of Tki5 for the α-helical toxic domain. Expression of constructs MIX-α-β$^{1-996}$ or MIX-α$^{1-863}$ that lack the PelB leader sequence is also toxic, although to a lesser extent than PelB-containing constructs. A schematic representation of the corresponding constructs is shown next to the corresponding plot. Plotted data are the average of three or six independent biological replicates ($n = 3, 6$). Error bars correspond to standard deviations (SD). Source data for this figure are available online. Source data are available online for this figure.

The sequence homology between Tap3 and Tap6 is strikingly low, at only ~ 16% (Appendix Fig. S11). Similarly, their corresponding cognate effectors, Tke5 and Ptx2, despite containing MIX domains, also share very low sequence identity (~17%; Appendix Fig. S11). The presence of Tap3-Loop$^{31-50}$, which is not present in Tap6 and accounts for over half of Tap3's binding residues to Tke5, marks a clear structural and mechanistic divergence in effector recognition. This divergence in these two Tap proteins likely reflects differences in how they specifically recognise their cognate effectors. Although both Tap3 and Tap6 recognise MIX-containing BTH_I2691 effectors, the distinct classification of Tke5 containing a Clan II MIX domain and Ptx2 containing a Clan I MIX domain further underscores the need for specialised adaptor recognition. This remarkable evolutionary plasticity suggests DUF4123 adaptors employ diverse structural strategies to specifically recognise a broad repertoire of effectors, thereby expanding the functional reach of the T6SS.

The challenges in computationally modelling the Tap6-Ptx2 interaction were mirrored by experimental hurdles. In a recent study, Colautti and colleagues attempted to determine the structure of the Tap6−Ptx2 complex using cryo-EM. These efforts were ultimately unsuccessful in yielding a 3D reconstruction, a difficulty the authors attributed to the low number of particles forming the complex and significant flexibility at the interaction site (Colautti et al, 2025). Nevertheless, their analysis of 2D class averages did reveal additional density near the N-terminal MIX domain of Ptx2, consistent with Tap6 binding at this location. These prior challenges highlight the significance of our Tap3-Tke5 structure, which successfully overcomes these obstacles to provide the first atomic-level visualisation of an adaptor-effector complex from this widespread family.

The recent cryo-EM structure of Ptx2 provides the first opportunity for a direct structural comparison between two members of the BTH_I2691 family (Colautti et al, 2025). A structural alignment of our Tke5 structure with that of Ptx2 reveals

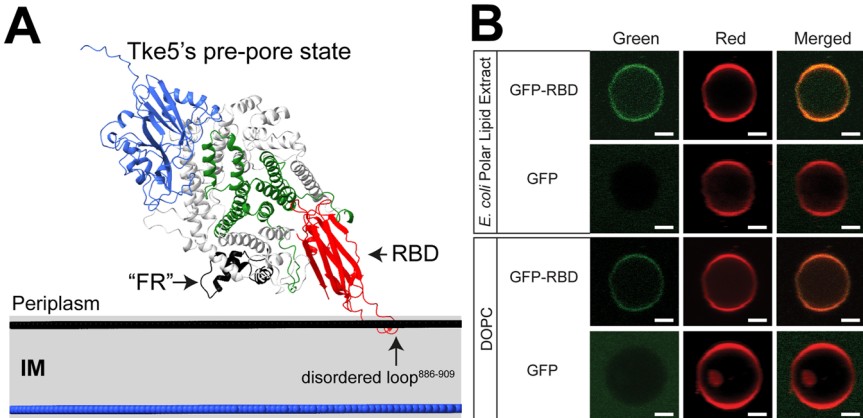

**Figure 6. The C-terminal β-rich region of Tke5 binds to model membranes in vitro.**

(A) Model of Tke5 interaction with the inner membrane of a target bacterium. Following delivery into the periplasmic space via the T6SS, Tke5 is hypothesised to initially interact with the inner membrane through its C-terminal β-rich receptor-binding domain (RBD, residues 864–996, shown in red). This initial binding, predicted by PPM 3.0, occurs via the distal end of the RBD, including a modelled disordered loop (residues 886–909). The "FR" label points to a region in Tke5 (residues 319–365, shown in black) that, in the homologue Ptx2, was proposed to function as a membrane sensor. (B) Binding of GFP–RBD to Giant Unilamellar Vesicles (GUVs). GUVs composed of *E. coli* polar lipid extract or pure DOPC were labelled with rhodamine-DHPE (red channel) and incubated with either GFP–RBD or GFP alone (green channel). Images show the GFP, rhodamine, and merged fluorescence signals. GFP–RBD colocalises with the vesicle membrane, whereas GFP signal shows no detectable membrane binding. Scale bars correspond to 2 μm. Source data for this figure are available online. Source data are available online for this figure.

that while the overall domain organisation is conserved, the proteins are remarkably divergent (Fig. 7A). Both effectors consist of an N-terminal MIX domain, a large central α-helical region that contains a predicted transmembrane domain (TMD), and a C-terminal β-rich region with an immunoglobulin-like fold. Despite this shared architecture, the overall structures differ significantly, with a root-mean-square deviation (RMSD) across aligned Cα atom pairs of 25 Å. This divergence is evident across all domains; when structurally aligned, the MIX domains, the α-helical regions, and the C-terminal β-rich regions show marked differences, with RMSD values of 6.2, 22.5, and 21.9 Å, respectively (Appendix Fig. S12). In Ptx2, the large α-helical scaffold domain forms a "claw-like" structure that completely encloses its TMD, a feature that is broadly consistent with the architecture of Tke5's α-helical region.

## Proposed Tke5's pore-forming molecular mechanism

The comprehensive analysis of Tke5's structure and the functional dissection of each domain allow for the proposal of a molecular mechanism for its pore-forming activity. Tke5 might operate through a sophisticated, multi-step process, culminating, as previously demonstrated, in the disruption of target bacterial membrane potential without causing widespread cellular lysis (Velázquez et al, 2025).

The proposed mechanism begins once Tke5 is delivered into the periplasm of a target bacterium, where it must interact with the inner membrane to exert its toxic effect. We hypothesise that this initial membrane interaction is mediated by the C-terminal β-rich region (residues 864–996), which adopts an immunoglobulin-like fold and functions as a putative Receptor-Binding Domain (RBD).

This model is supported by three key observations. It is consistent with previous in vitro electrophysiological data showing that Tke5 senses the directionality of the membrane potential and

preferentially inserts into the inner membrane from the periplasmic side (Velázquez et al, 2025). Furthermore, our functional assays demonstrate that while the RBD is not essential for toxicity (Fig. 5C,D), it is capable of binding directly to model membranes (Fig. 6B). The RBD's membrane-sensing role is therefore likely guiding Tke5 to its target cytoplasmic membrane and orienting it for the subsequent insertion of the toxic domain. We cannot, however, rule out the possibility that the RBD also binds to a yet-to-be-discovered membrane protein receptor.

Following initial membrane sensing by the RBD, the α-helical region (residues 239–863) is proposed to undergo a significant conformational change to form the pore. In the soluble, pre-pore state, the five segments with a strong propensity for membrane insertion are either buried within the protein's core or located on its surface (Fig. 6A). Computational modelling with AlphaFold3 (AF3; (Abramson et al, 2024)) predicts that upon membrane interaction, this region undergoes an extensive conformational rearrangement (Fig. 7B; Appendix Fig. S13). The AF3 model, combined with PPM 3.0 calculations, suggests these helices refold to form a transmembrane domain (TMD) that crosses the lipid bilayer. Taken together, these predictions support a multi-step mechanism: initial membrane recognition by the C-terminal RBD tethers Tke5 to the membrane, facilitating a large-scale conformational rearrangement of the α-helical region that leads to the insertion of the TMD and subsequent pore formation.

The recent cryo-EM structure of *P. aeruginosa* Ptx2 revealed a "Flexible Region" (FR) within its α-helical domain, proposed to function as a membrane sensor that initiates a conformational change upon contact with the lipid bilayer, thereby exposing its transmembrane domain (TMD). Critically, this FR was not resolved in the Ptx2 cryo-EM density, a fact attributed to a high degree of intrinsic flexibility consistent with its proposed role as a dynamic sensor.

Notably, the corresponding region in Tke5 (residues 319–365) is well-resolved in our cryo-EM structure. In our model of Tke5's pre-

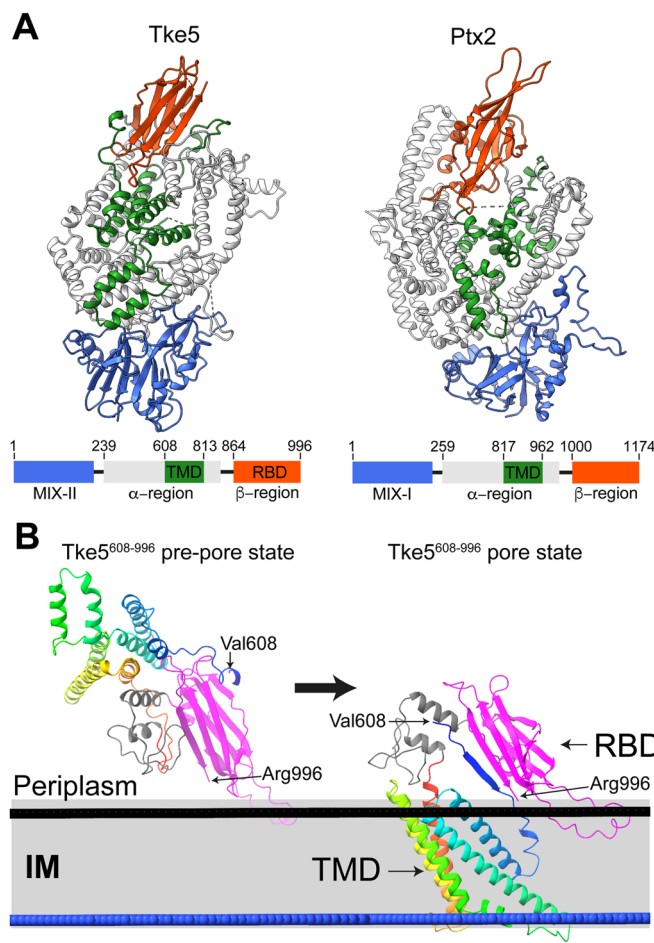

**Figure 7. Side-by-side comparison of Tke5 and Ptx2 and proposed molecular mechanism for Tke5 pore formation.**

(A) Side-by-side comparison of Tke5 and Ptx2 showing their overall architecture. They contain an N-terminal MIX domain classified as clan II and I for Tke5 and Ptx2, respectively (shown in blue), a central α-helical region that we have demonstrated is sufficient for toxicity in Tke5 (shown in grey and green, where green indicates predicted transmembrane helices), and a C-terminal β-rich region (shown in red) that we show for Tke5 is not necessary for toxicity, but can bind to model membranes. The domain boundaries for Tke5 and Ptx2 are also shown below their corresponding structures. (B) Proposed molecular mechanism for Tke5 pore formation. Predicted conformational change of Tke5's TMD upon membrane interaction. A comparison between the cryo-EM structure of the Tke5 C-terminal region (residues 608–996) and the AlphaFold3 (AF3) predicted structure for the same residue range suggests that predicted transmembrane helices in Tke5 (shown with a rainbow colour scheme) undergo an extensive conformational change upon membrane interaction. Combined with PPM 3.0 calculations, these predictions suggest that TM helices cross the lipid bilayer, forming a Transmembrane Domain (TMD) that is proposed to be responsible for pore formation upon membrane binding. This multi-step process disrupts the target bacterial membrane potential. PPM 3.0 calculations are performed with a model *E. coli* inner membrane (PE:PG:CL (79:19:2)). The membrane is shown schematically in a perpendicular orientation, with black and blue surfaces representing the interface between the lipid polar head groups and the acyl chains. The Tke5 N-terminal region (residues 1–607), which includes the MIX domain, is not shown for clarity. Source data for this figure are available online. Source data are available online for this figure.

pore state (Fig. 6A), the "FR" region is positioned close to the membrane surface. This suggests that, analogous to what has been proposed for Ptx2 (Colautti et al, 2025), this region of Tke5 might also be involved in the large-scale conformational rearrangement of

the α-helical region that leads to the insertion of the TMD and subsequent pore formation.

Further illustrating the family's adaptability, the recent characterisation of the divergent homologue Bte2 from *Bacteroides fragilis* revealed a highly specialised, receptor-dependent mechanism (Ratner et al, 2025). Ratner et al, demonstrated that Bte2 intoxication is strictly dependent on the presence of the Rnf electron transport complex in the target cell's inner membrane. Crucially, the catalytic activity of the Rnf complex was shown to be dispensable for toxicity, confirming that Bte2 co-opts the complex as a physical receptor to facilitate its membrane insertion. Bte2 also contains a C-terminal immunoglobulin-like fold (Appendix Fig. S11), but currently, no experimental data show which region of Bte2 is involved in binding to the Rnf receptor.

Together, these findings suggest that the BTH_I2691 family employs a modular, evolutionarily adaptable mechanism for membrane insertion, likely involving two key regions: an internal "Flexible Region" (FR), as described for Ptx2, and a C-terminal Receptor-Binding Domain (RBD), as observed in Tke5. We propose a unified model in which the FR-like region serves as the primary membrane-sensing and insertion-triggering module, consistent with findings that this region is essential for Ptx2's toxicity.

The C-terminal RBD likely functions as an accessory module that provides an enhanced tethering mechanism. In toxins like Tke5, the RBD can mediate an initial, high-avidity interaction with the membrane, thereby increasing the toxin's local concentration and positioning the FR-like region to efficiently trigger the conformational changes required for pore formation. This two-tiered model provides a framework that integrates available functional data. The fact that the Tke5 RBD is dispensable for toxicity when expressed in the cytoplasm of sensitive bacteria suggests that the FR-like region can independently initiate membrane insertion. In contrast, the RBD's demonstrated ability to bind membranes in vitro supports its role in enhancing this process. This modularity allows for a spectrum of targeting strategies within the family, from generalist membrane-sensing driven primarily by the FR to more specialised, RBD-assisted targeting.

## Electrophysiological properties and diverse mechanisms of T6SS Pore-Forming Toxins causing bacterial cell depolarisation

The T6SS deploys a diverse arsenal of toxic effector proteins, including PFTs such as VasX (Miyata et al, 2013), Tme1 (Fridman et al, 2020), Tme2 (Fridman et al, 2020), Ptx2 (Colautti et al, 2024), Tse5 (Rojas-Palomino et al, 2025a; González-Magaña et al, 2022), Tse4 (Rojas-Palomino et al, 2025b), Ssp4 (Reglinski et al, 2025), Ssp6 (Mariano et al, 2019), and Bte2 (Ratner et al, 2025). The unifying principle among these PFTs is their ability to induce membrane depolarisation as a primary mechanism of bacterial intoxication. The electrophysiologic properties of Tse5 (Rojas-Palomino et al, 2025a; González-Magaña et al, 2022), Tse4 (Rojas-Palomino et al, 2025b), Ssp4 (Reglinski et al, 2025), and Ssp6 (Mariano et al, 2019), as well as Tke5 (Velázquez et al, 2025), have been studied in detail, demonstrating that their toxicity is due to their capacity to assemble ion-selective pores in the inner membrane of Gram-negative bacteria, generally preferring cations over anions. Nonetheless, their specific biophysical properties and modes of action exhibit remarkable diversity (Table 2).

**Table 2. Comparative electrophysiological analysis of Tke5, Tse4, Tse5, Ssp4, and Ssp6.**

| Characteristic | Tke5 (Velázquez et al, 2025) | Tse4 (Rojas-Palomino et al, 2025b) | Tse5 (González-Magaña et al, 2022; Rojas-Palomino et al, 2025a) | Ssp4 (Reglinski et al, 2025) | Ssp6 (Mariano et al, 2019) |
|---|---|---|---|---|---|
| Source organism | *P. putida* KT2440 | *P. aeruginosa* PAO1 | *P. aeruginosa* PAO1 | *Serratia marcescens Db10* | *Serratia marcescens Db10* |
| Toxin family/type | BTH_I2691 family PFT | T6SS effector PFT | Rhs toxin, T6SS PFT | Ssp4-like proteins | [a]Tpe1 |
| Primary mechanism of action | Membrane depolarisation | Membrane depolarisation | Membrane depolarisation | Membrane depolarisation | Membrane depolarisation |
| Membrane damage | No large damage | No large damage | No large damage | No large damage | No large damage |
| Pore conductance | [b]$33 \pm 4$ pS | [b]$20 \pm 3$ pS | [b]$196 \pm 8$ pS[c] | [d]$18.4 \pm 0.64$ pS | ND |
| Estimated pore radius (nm) | ~0.4 | ≤0.4 | ~0.9 | ND | ND |
| Reversal potential (mV) | [e]$-31 \pm 2$ | [e]$-20$ (variable) | [e]$-14.0 \pm 2.1$ | [f]$-14.8 \pm 1.6$ | [f]$-55.8 \pm 3.4$ |
| Ion permeability ratio ($P_{K+}/P_{Cl}^-$) | $10 \pm 3$ | $4.0 \pm 1.9$ | $3.8 \pm 0.9$ | ND | ND |
| Ion selectivity | Cation-selective | Cation-selective | Cation-selective | Cation-selective | Cation-selective |
| Specific cation preference | ND | $Na+$, $Li+ > K+$ | No cation specificity | No cation specificity | [g]$K^+ \approx Na^+$ |
| Voltage dependence/ rectification | Ohmic | Ohmic (NaCl, KCl) & Outward Rectifying (KCl) | Initial voltage-dependent, then Ohmic | Ohmic | ND |
| Proposed insertion mechanism | Periplasmic insertion | Periplasmic insertion | Periplasmic insertion | Periplasmic insertion | Periplasmic insertion |
| Immunity protein | Tki5 | Tsi4 | Tsi5 | Sip4 | Sip6 |

*ND* not determined.

[a]T6SS-dependent pore-forming effector.

[b]Measured in symmetrical 150/150 mM KCl (cis/trans chambers) and lipid membrane assembled from *E. coli* polar lipid extract.

[c]Unpublished data.

[d]Measured in symmetrical 510 mM KCl and a membrane assembled from bovine phosphatidylethanolamine lipids.

[e]Measured in asymmetrical 250/50 mM KCl (cis/trans chambers) and lipid membrane assembled from *E. coli* polar lipid extract.

[f]Measured in asymmetrical 500/210 mM KCl (cis/trans chambers) and lipid membrane assembled from bovine phosphatidylethanolamine lipid.

[g]Measured in bi-ionic conditions (210 mM NaCl/210 mM KCl in cis/trans chambers).

Tke5 forms subnanometric (~0.4 nm radius) ohmic pores with some preference for cations over anions ($P_K^+/P_{Cl}^- = 10 \pm 3$), primarily causing depolarisation via $Na^+$ influx (Velázquez et al, 2025). Tse4, which also forms narrow pores (with a radius of ≤ 0.4 nm), uniquely combines ohmic and diode-like rectifying channels (Rojas-Palomino et al, 2025b). Thus, Tse4 exhibits an exquisite mechanism of action whereby initial depolarisation is postulated to result from the influx of $Na^+$ through Tse4-assembled ohmic pores that operate under a resting membrane potential (Rojas-Palomino et al, 2025b). These pores show mild cation selectivity for $Na^+$ and $Li^+$ over $K^+$ ($P_K^+/P_{Cl}^- = 4.0 \pm 1.9$). After cell depolarisation (i.e., at positive membrane potentials), Tse4-rectifying pores activate to facilitate $K^+$ efflux, thereby balancing charge without altering pH (LaCourse et al, 2018). Tse5 forms nanometric proteolipidic pores (~0.9 nm radius) that initially exhibit voltage-dependent conductance before stabilising into ohmic behaviour (González-Magaña et al, 2022; Rojas-Palomino et al, 2025a). Its weak cation selectivity ($P_K^+/P_{Cl}^- = 3.81 \pm 0.86$) is significantly influenced by membrane lipid composition (Rojas-Palomino et al, 2025a).

Ssp4 forms pores ($18.4 \pm 0.64$ pS conductance) that also showed cation selectivity. Notably, Ssp4 intoxication leads to an increase in intracellular reactive oxygen species (ROS) (Reglinski et al, 2025). Ssp6, another cation-selective PFT, shows a preference for monovalent cations ($Na^+$, $K^+$) over divalent ones ($Ca^{2+}$) and is not permeable to protons (Mariano et al, 2019).

Despite their shared outcome of membrane depolarisation and non-lytic growth inhibition, these subtle differences in biophysical characteristics, such as voltage-dependent rectification and specific cation selectivity, highlight the evolutionary fine-tuning of these PFTs to achieve specific physiological outcomes within the target cell. Tse4's rectifying behaviour may enable more controlled $K^+$ efflux, potentially synergising with other effectors or adapting to specific intracellular conditions. At the same time, Tke5's mechanism appears more straightforward in its cation influx-driven depolarisation. Tse5's unique encapsulation and release mechanism, along with its strong lipid dependence, further illustrate the diverse strategies employed by these toxins. This convergence on the inner membrane as a target underscores its fundamental importance in bacterial physiology, suggesting that PFTs that specifically and efficiently disrupt its electrochemical potential represent a highly successful evolutionary strategy for interbacterial competition.

## Summary

Our study elucidates the structural and functional basis of *Pseudomonas putida* Tke5, a pore-forming toxin that antagonises plant pathogens. The cryo-EM structure of the Tap3–Tke5 complex at 2.8 Å resolution reveals two novel protein folds, providing the

first atomic-level insight into how MIX motifs mediate effector-adaptor interactions. We demonstrate that Tap3, a DUF4123 family adaptor, specifically recognises Tke5. Functional dissection identifies the α-helical region of Tke5 as the core pore-forming module responsible for membrane depolarisation and bactericidal activity. While the C-terminal β-rich region likely enhances target membrane specificity. Crucially, the immunity protein Tki5 directly neutralises the α-helical toxicity, underscoring the evolutionary precision of effector-immunity pairing. These findings establish a model for MIX-dependent effector recruitment, emphasising the modular architecture of T6SS toxins. By integrating structural and functional analyses, this work advances our understanding of bacterial competition mechanisms and provides significant insight into the widespread BTH_I2691 protein family.

A key question for future studies is the precise stoichiometry and architecture of the functional Tke5 pore. While our data clearly demonstrate pore-forming activity leading to selective ion transport, the exact number of subunits required to form the pore remains to be determined. Elucidating the pore's structure will require dedicated biophysical approaches, such as cryo-EM, and represents an exciting avenue for future investigation into the mechanism of the BTH_I2691 toxin family.

# Methods

### Reagents and tools table

| Reagent/resource | Reference or source | Identifier or catalogue number |
|---|---|---|
| **Experimental models** | | |
| *Escherichia coli* BL21(DE3): F-, ompT, hsdSB(rB–mB–), gal, Ion, λ(DE3 [lacI lacUV5-T7p07 ind1 sam7 nin5]) [malB + ]K- 12(λS), dcm | Sigma-aldrich | 69450-M |
| *Pseudomonas putida* KT2440R: wild-type strain, RifR | Espinosa-Urgel et al, 2000 | |
| **Recombinant DNA** | | |
| **pJET1.2/blunt** | Thermo Scientific | K1232 |
| **pS238D•M** | Calles et al, 2019 | |
| **pS238D•tke5** | Velázquez et al, 2025 | |
| **pS238D•pelB-tke5** | Velázquez et al, 2025 | |
| **pS238D•pelB-tke5_β** | This work | |
| **pS238D•pelB-tke5_MIX** | This work | |
| **pS238D•pelB-tke5_α + β** | This work | |
| **pS238D•pelB-tke5_α** | This work | |
| **pS238D•tke5_MIX + α** | This work | |
| **pSEVA621** | Silva-Rocha et al, 2013 | |
| **pSEVA234C** | Nikel et al, 2022 | |
| **pSEVA624C** | Velázquez et al, 2025 | |
| **pSEVA424** | Silva-Rocha et al, 2013; Martínez-García et al, 2023a, 2023b | |
| **pSEVA424•tki5** | Velázquez et al, 2025 | |
| **pSEVA624C•tki5** | Velázquez et al, 2025 | |
| **pCOLADuet™-1** | Sigma-Aldrich | 71406-M |
| **pCOLADuet-1•tke5** | This work | |
| **pCOLADuet-1•6xhis-tap3-tke5** | This work | |
| **pET29a(+)•9xhis-gfp-tke5_β** | This work | |

| Reagent/resource | Reference or source | Identifier or catalogue number |
|---|---|---|
| **Oligonucleotides and other sequence-based reagents** | | |
| P1: aattaaGCTAGCactaacgcctccgtaagcag | This work | NheI.tke5.F/ pS238D•M |
| P2: ggccatGGATCCgtttagcgttc cagatgaatagtgg | This work | BamHI.tke5.R/ pS238D•M |
| P3: aattaaGCTAGCaagtacctgctgc cgaccgccgccgccggcctgctgctgctggccgc ccagccggccatggccactaacgcctccgtaagcag | This work | NheI.pelBtke5.F/ pS238D•M |
| P4: atacGCTAGCaagtacctgctgccg ccgccgccgccggcctgctgctgctggcc gcccagccggccatggccTCACCGAAAG CATTGTTGGCACC | This work | NheI.RBD.F/ pS238D•M |
| P5: ATATATggatcCttaggcgagcgccaccaccac | This work | BamHI.MIX.R/ pS238D•M |
| P6: TATTAAgCTAGCaagtacctgctgccgacc gccgccgccggcctgctgctgctggccgcccagccggccatg gccgacgccgaaggcatggccctg | This work | NheI.ALPHA.F/ pS238D•M |
| P7: tagaGGATCCttagaaaaattc aatctgccatgacttgatttcttcg | This work | NheI.ALPHA.R/ pS238D•M |
| P8: tgctgcaactctctcaggg | This work | S238D-F |
| P9: gggttttcccagtcacgac | This work | S238D-R |
| P10: agcggataacaatttcacacagga | This work | SEVA-F |
| P11: cgccagggtgtttcccagtcacgac | This work | SEVA-R |
| **Chemicals, enzymes, and other reagents** | | |
| Isopropyl-β-ᴅ-thiogalactopyranoside (IPTG) | Apollo Scientific | BIMB 1008 |
| LB Miller | Teknova | 621551.05 |
| Kanamycin Monosulfate | Apollo Scientific | BIK0126 |
| Glucose | Merck | G8270-1KG |
| Gentamycin sulphate | Apollo Scientific | BIG0124 |
| 3-methylbenzoate (3-*m*Bz) | Thermo Fisher | A13785 |
| SurePAGE™, Bis-Tris,10×8 | Genscript | M00656 |
| 70Ti Fixed-Angle Titanium Rotor | Beckman Coulter | 337922 |
| Benzonase® Nuclease, Purity > 90% | Merck | 70746-3 |
| cOmplete, EDTA-free Protease Inhibitor Tablets | Merck | 5056489001 |
| Imidazole | Merck | 1047161000 |
| DTT (Dithiothreitol) (>99% pure) Protease free | Apollo Scientific | BIMB1005 |
| Amicon Ultra-15, membrana PLTK Ultracel-PL, 30 kDa | Fisher Scientific | 11829660 |
| Filtropur S µm | Sarstedt | 831.826.001 |
| HiLoad® 26/600 Superdex® 200 pg | Cytiva | 28-9893-36 |
| HisTrap™ High Performance 5 ml | Cytiva | 17-5248-02 |
| Superdex™ 200 Increase 10/300 GL | Cytiva | 28-9909-44 |
| 3-((3-colamidopropilo) dimetilamonio)-1-propanesulfonato (CHAPS) | Merck | C5070-5G |
| Quantifoil™ R 1.2/1.3 300 mesh grids | Jena Bioscience | X-101-CU300 |
| ELMO Glow discharge Clean | Agar Scientific | GDS203501 |
| Leica EM GP2 | Leica Microsystems | 611569 |
| *E. coli* polar lipid extract | Avanti Polar Lipids | 100600 C |
| 1,2-dioleoyl-*sn*-glycero-3-phosphocholine (DOPC) | Avanti Polar Lipids | 850375 P |
| Rhodamine B 1,2-dihexadecanoyl-*sn*-glycero-3-phosphoethanolamine triethylammonium salt | Invitrogen | L1392 |
| ITO-coating (≤ 10 Ω/sq.) coated on selected float glass substrate | Präzisions Glas & Optik GmbH | CEC010S |
| µ-Slide 8 well high glass bottom chambers | Ibidi | AI-80827 |
| **Software** | | |
| Prism 9 version 9.5.1 | GraphPad | |

| Reagent/resource | Reference or source | Identifier or catalogue number |
|---|---|---|
| CryoSPARC v4.5.3 | Structura Biotechnology Inc. | |
| **Other** | | |
| Krios G4 Cryo-EM | ThermoFisher | |

## Construction of plasmids and bacterial strains

For a detailed overview of all strains and plasmids used in this study, please refer to the Reagents and Tools Table. Plasmid DNA sequences are available in Dataset EV1. All cloning procedures were conducted as described in previous work (Velázquez et al, 2025). In brief, the gene coding for Tke5 (PP2612) was amplified from genomic DNA isolated from the *Pseudomonas putida* KT2440 strain using P1-P2 primers. Additionally, P2-P3 primers were employed to amplify Tke5 and fuse a PelB signal sequence at the 5′-end. The *tke5* and *pelB·tke5* genes were first cloned into pJET1.2/blunt (ThermoFisher Scientific™) and verified by Sanger sequencing using Macrogen Inc. services. Subsequently, *tke5* and *pelB·tke5* were subcloned into the pS238D•M vector at NheI/BamHI sites, resulting in pS238D•*tke5* and pS238D•*pelB·tke5*, respectively. This broad host range and medium copy number vector originates from the SEVA collection (Martínez-García et al, 2023a).

Then, pS238D•M was used as the parental vector to generate a series of plasmids containing specific Tke5 domains fused to the N-terminal signal sequence. Tke5_β (864-996) was amplified using primers P4-P2 and then cloned into pS238D•M to generate pS238D•*pelB· M tke5_β*. The Tke5-MIX domain (1–238) was amplified using primers P3 and P5 and then cloned into pS238D•M to generate pS238D•*pelB·tke5_MIX*. The coding DNA region for Tke5_α + β (239–996) was amplified using primers P6 and P2 and then cloned into pS238D•M to generate pS238D•*pelB·tke5_α + β*. Tke5_α (239–863) was amplified using primers P6-P7 and then cloned into pS238D•M to generate pS238D•*pelB·tke5_α*. Additionally, pS238D•M was also used as the parental vector to generate the plasmid pS238D•*tke5_MIX*-α (1–863). To do so, Tke5_MIX domain + α (1–863) was amplified using primers P1 and P7 and then cloned into pS238D•M. Colony PCR using primers P8-P9 were used to screen colonies for pS238D•M derivatives and their subsequent sequencing.

Cloning of the *tki5* (PP2611) gene was performed as described in previous work (Velázquez et al, 2025). In brief, the *tki5* (PP2611) gene with an artificial RBS region (TTTAAAGGAGATATACAA) at the 5'-end was synthesised and cloned between the KpnI and HindIII restriction sites into the pSEVA424 vector by GenScript (Reagents and Tools Table). Then, both the RBS and *tki5* were subcloned from pSEVA424•*tki5* into pSEVA624C at KpnI/HindIII sites to generate pSEVA624C•*tki5*. This vector is a derivative of pSEVA621 that comes from the SEVA collection (Silva-Rocha et al, 2013; Martínez-García et al, 2023b) and was generated in a previous work where *lacI*q-P*trc* cargo from pSEVA234C (Nikel et al, 2022) was inserted in the MCS using PacI/HindIII (Velázquez et al, 2025). By colony PCR using P10-P11 primers, colonies harbouring the insert *tki5* were screened and sequenced.

The Tke5 coding sequence was subcloned from pS238D•*tke5* into the multiple cloning site 2 (*mcs-2*) of pCOLADuet™-1. This

subcloning was performed by GenScript (GenScript, New Jersey, USA), resulting in the plasmid pCOLADuet-1•*tke5*. The *tap3* (PP2613) gene was synthesised and cloned into the *MCS-1* of pCOLADuet-1•*tke5* by using the GenScript service. This subcloning conserves the 5′ 6xHis-tag of the *mcs-1* and leads to pCOLADuet-1•6x*his-tap3-tke5* (Reagents and Tools Table).

## Expression, purification and cryo-EM vitrification of Tap3–Tke5 complex

*Escherichia coli* Bl21(DE3) cells co-transformed with pCOLADuet-1•6x*his-tap3-tke5* and pSEVA624C•*tki5* plasmids were grown overnight at 37 °C with constant agitation in 200 mL of LB media containing 50 μg mL$^{-1}$ kanamycin (Km), 20 μg mL$^{-1}$ gentamycin (Gm) and 0.2% glucose. For Tap3 and Tke5 overexpression, bacterial cultures were diluted to an initial $OD_{600}$ of 0.1 in flasks with 2 L of fresh LB medium supplemented with Km and Gm at the same concentrations but lacking glucose, and then they were allowed to grow at 37 °C with shaking conditions. In the absence of glucose, Tki5 is expressed to neutralise possible Tke5-induced toxicity. When cells reached an $OD_{600}$ value of 0.7, protein expression was induced by adding isopropyl β-D-1-thiogalactopyranoside (IPTG) at a final concentration of 1 mM. The temperature was reduced to 18 °C, and bacterial cultures were incubated overnight with agitation. Cells were harvested by centrifugation at 6000×$g$ for 20 min, and pellets were stored at −80 °C for later use.

The pellet from 4 L of bacterial culture was resuspended in 60 mL of solution A (50 mM Tris-HCl, pH 8.0, 500 mM NaCl, 20 mM imidazole) with 5 μL of benzonase endonuclease (Millipore, Sigma) and a tablet of protease inhibitor cocktail (cOmplete, EDTA-free, Roche). Continuous cycles of 15 s ON and 59 s OFF at 60% amplitude were performed for 5 min to lyse cells by sonication. The bacterial lysate was ultra-centrifuged at 125,748×$g$ for 1 h. The soluble fraction was first passed through a 0.2 μm filter and then loaded using a peristaltic pump into a 5 mL HisTrap HP column equilibrated with solution A.

The column was connected to a fast protein liquid chromatography system (ÄKTA FPLC; GE Healthcare) and washed with solution A at 1 mL min$^{-1}$. When the absorbance at 280 nm stabilised near the baseline, the Tap3–Tke5 complex was eluted with 100% solution B (50 mM Tris-HCl, pH 8, 500 mM NaCl and 500 mM imidazole) at 4 mL min$^{-1}$. Fractions of the central peak were pulled, and dithiothreitol (DTT) was added at a final concentration of 2 mM. The sample was then injected into a HiLoad Superdex 200 26/600 pg column, previously equilibrated with solution C (20 mM Tris-HCl, pH 8, 150 mM NaCl and 2 mM DTT). The Tap3–Tke5 complex eluted as a single, monodispersed peak, and SDS-PAGE gel analysis showed two bands (Fig. 1B), which were confirmed by mass spectrometry to correspond to Tke5 and Tap3 (Dataset EV1). Central peak fractions corresponding to the complex were collected and concentrated using an Amicon centrifugal filter unit with a 10 kDa molecular mass cut-off (Millipore) to a final concentration of 9 mg mL$^{-1}$ (~1.5 mg of protein is obtained for each litre of culture).

About 0.5 mL of the freshly concentrated sample was injected into a Superdex™ 200 Increase 10/300 GL, previously equilibrated with solution C. Fractions of 50 μl were collected. The elution peak at 2.2 mg mL$^{-1}$ was used to prepare grids after adding 0.05% CHAPS to the sample 5 min before vitrification, which is essential

to avoid the preferred orientation pathology (Kampjut et al, 2021). Quantifoil™ R 1.2/1.3 300 mesh grids were glow-discharged at a 0.41 mbar vacuum for 90 s. Then, 4 µL of purified Tap3–Tke5 at 2.2 mg mL⁻¹ in solution C with 0.05% CHAPS were applied to the grids and blotted for 2.3 s in a Leica EM GP2 single-side blotting automated plunge freezer at 95% humidity. Grids were plunge-frozen in liquid ethane and stored in liquid nitrogen until data collection.

## Cryo-EM data collection and processing

### Cryo-EM grid preparation

Grids were clipped in liquid nitrogen and were introduced into an in-house 300 kV Thermo-Fisher Titan Krios G4 transmission electron microscope, where data were collected paired with Gatan's BioContinuum Imaging Filter. In total, 19,112 movies were recorded on a K3 direct electron detector device at a nominal magnification of ×130,000 with a calibrated pixel size of 0.6462 Å. A defocus range of −0.8 to −2.5 µm was used, with a total dose of 48.8 e − /Å² fractionated over 50 frames with a total exposure time of 1.3 s. Acquired image stacks were processed using CryoSPARC (Punjani et al, 2017) software, resulting in a final map that allowed the building of the atomic structure of Tap3–Tke5 at 2.8 Å.

### Cryo-EM data processing

In total, 19,112 raw cryo-EM movies were preprocessed in CryoSPARC (Punjani et al, 2017) with Patch Motion Correction and Patch CTF Estimation. Template-based particle picking was performed with templates created from previous low-resolution reconstructions, followed by an initial 2D classification of 3× binned particles. Further rounds of 2D classification with varying parameters were used to improve particle quality and remove duplicates. The best 2D classes were re-extracted with no binning for an initial ab initio reconstruction into three classes. The best volume (referred to as the *initial volume*) was chosen for further 2D classification. From the best 2D classes, focused particle picking was then performed using Topaz (Bepler et al, 2020), with two separate models for different particle orientations, separating top/bottom views from side views. Particles were re-extracted with 3× binning (384 px box size, 128 px binning). Relaxed 2D classification criteria were applied to retain usable particles while removing the main unusable particles. Duplicates closer than 150 Å were removed when combining both particle sets. The final cleaning steps included the exclusion of aggregates. The initial volume was used as an input for Non-Uniform (NU) refinement (Punjani et al, 2020) with the latest particle set. A new ab initio reconstruction and heterogeneous refinement cycles were performed, in which the NU refinement volume coming from the initial volume was included as an input with the rest of the volumes from the ab initio reconstruction, and the best-resolved class was chosen for particle extraction with no binning and a box size of 384 px. A last heterogeneous refinement step was performed before re-extracting the particles in the best volume with a bigger box (600 px). Post-processing included Local CTF Refinement and Reference-Based Motion Correction. A final NU refinement was performed, achieving a 2.8 Å resolution. CryoSPARC (Punjani et al, 2017) Orientation Diagnostics was used to calculate the Sampling Compensation Factor (SCF*) and the corrected Fourier Amplitude Ratio (cFAR). SCF* and cFAR values of 0.827 and 0.57 indicate no particle orientation bias or map anisotropy. The final 2.8 Å density map was enhanced with DeepEMhancer (Sanchez-Garcia et al, 2021) for model building and visualisation. Cryo-EM data processing

workflow and Cryo-EM maps and data quality are shown in Appendix Figs. S1–S3. Data collection and processing statistics are provided in Table 1.

## Model building and structure refinement

Approximately 65% of the Tke5 structure and 89% of the Tap3 structure were reconstructed using automatic ab initio model building with ModelAngelo (Jamali et al, 2024) and the density-modified DeepEMhancer (Sanchez-Garcia et al, 2021) cryo-EM map. The remaining regions were completed through multiple iterative rounds of manual model building performed in Coot (v0.9.8.91) (Casañal et al, 2020) and real-space refinement in Phenix (1.20.1-4487 package) (Liebschner et al, 2019; Afonine et al, 2018) against the final NU refinement cryo-EM map. The final structure contains 94% and 92% of Tap3 and Tke5 sequences, respectively. Secondary structure and Ramachandran restraints were included in the refinement. MolProbity (Williams et al, 2018) and the PDB validation server (Gore et al, 2017) were used for model validation. A summary of the final model statistics is provided in Table 1.

## Growth inhibition and immunity-driven recovery assays

*P. putida* KT2440 cells were electroporated with two compatible plasmids, pS238D•M and pSEVA624C, or their derivatives harbouring *pelB·tke5, pelB·tke5_β, pelB·tke5_MIX, pelB·tke5_α + β, pelB·tke5_α, tke5* or *tke5_MIX-α*, and *tki5*, respectively, as described in the Reagents and Tools Table. Transformed cells were selected on LB-agar plates supplemented with Km 50 µg mL⁻¹ (for pS238D•M) and Gm 25 µg mL⁻¹ (for pSEVA624C). The presence of the plasmid was confirmed by colony PCR using primers P8-P9 for pS238D•M derivatives and P10-P11 for pS624C derivatives, as listed in the Reagents and Tools Table.

For toxicity assays, overnight cultures of each strain were grown in an LB medium supplemented with the corresponding antibiotics. The following day, cultures were diluted to an $OD_{600}$ of 0.05 in fresh medium, and 1 mM IPTG was added to induce *tki5*. Aliquots of 200 µL were added to a microtiter plate, and $OD_{600}$ values were measured every 15 min at 30 °C using the Synergy/H1 microplate reader (Biotek). After an initial two-hour incubation, 0.5 mM of 3-methylbenzoate (3-*m*Bz) was added to each well and cells were incubated for another 12 h. Data were collected for three biological replicates, each with technical duplicates.

## In vitro visualisation of Tke5_β–membrane interaction by fluorescence microscopy

All lipids were obtained from Avanti Polar Lipids (see "Methods" for details). Lipid stocks were prepared from chloroform solutions using either *Escherichia coli* polar lipid extract or pure 1,2-dioleoyl-*sn*-glycero-3-phosphocholine (DOPC), and diluted to a final concentration of 1 mg mL⁻¹. Rhodamine B 1,2-dihexadecanoyl-*sn*-glycero-3-phosphoethanolamine triethylammonium salt (rhoda-mine-DHPE) (Invitrogen™ Lissamine™) was incorporated into each lipid mixture at 0.5%. All buffers were adjusted to be isosmotic and were equilibrated to room temperature before the experiment.

Tke5^864-996 (Tke5_β) was fused to a GFP protein at its N-terminal region together with an N-terminal 9xhis-tag and cloned into a

pET29a(+) plasmid by GenScript (pET29a(+)•*9xhis-gfp-tke5_β*; see Dataset EV1 for details). Protein expression and purification of GFP-Tke5_β were performed as described for the Tap3-Tke5 complex (see "Methods" for more information). The GFP protein was kindly provided by another laboratory at the *Instituto Biofisika* (CSIC, EHU) and used as a negative control.

Giant unilamellar vesicles (GUVs) were formed by electroformation on glass slides coated with indium tin oxide (ITO; sheet resistance ≤10 Ω/sq, thickness ≈1.1 mm, thin $SiO_2$ passivation layer between substrate and ITO, dimensions ≈25 mm × 75 mm) (Präzisions Glas & Optik GmbH).

For each lipid composition, 20 μL of the lipid solution was dropped onto a small, defined area on the conductive side of two ITO glass slides using a Hamilton microsyringe. Lipids were dried at 37 °C for 20 min under constant vacuum. Each pair of ITO glass slides was assembled with their conductive sides facing inward and separated only by an O-ring placed within the defined area. The slides were held securely together using a clamp to prevent leakage during GUV electroformation. Then, 300 mM sucrose solution was added inside the O-ring until it was filled, and the remaining upper opening was sealed with glue.

GUV electroformation was performed at 37 °C (Incubator Labnet 5110) by connecting a pair of electrodes, one electrode attached to each ITO glass slide, to a function generator (TG315) and applying an alternating electric field (10 Hz, 2.7 V) for 1 h 30 min. This resulted in the formation of vesicles containing sucrose.

Experiments were performed in μ-Slide 8-well high glass bottom chambers (Ibidi®), which were pre-blocked with 10 mg mL$^{-1}$ BSA for 15 min. Then, 80 μL of GUVs, protein at a final concentration of 400 nM, and the required volume of 150 mM KCl, 10 mM HEPES (pH 7.4) buffer were added to each well to reach a final volume of 400 μL. After a 30-min incubation, confocal fluorescence microscopy was performed using a Leica TCS SP5. Images of rhodamine-DHPE labelled GUVs were acquired by exciting at 543 nm and collecting from 550 to 615 nm. GFP was excited at 488 nm, and emission was collected from 500 to 540 nm.

### Bioinformatic and computational analysis

#### Tke5 topology prediction
The topology of Tke5 was predicted using the TMHMM-2.0 algorithm (DTU Health Tech (Krogh et al, 2001)) with the default settings.

#### Structural-based homology search of Tap3 and Tke5
The cryo-EM structures of Tke5 and Tap3 were used as a query to search for homologues using the Foldseek Search Tool (van Kempen et al, 2024).

#### Tke5 protein structure prediction and membrane positioning
The C-terminal residues (608–996) of Tke5 (Tke5$^{608-996}$) were modelled using AlphaFold 3 (Abramson et al, 2024). The disordered regions in the cryo-EM Tap3–Tke5 structure, including a loop within the putative receptor-binding domain (RBD) (residues 886–909), were modelled using AlphaFold_unmasked (Mirabello et al, 2024). The spatial orientation and optimal positioning of the Tke5 pre-pore state and the AlphaFold 3 prediction of Tke5$^{608-996}$ in a model Gram-negative inner membrane were predicted using the PPM 3.0 server (Protein Property

Prediction and Modelling Server (Lomize et al, 2022)). The model membrane composition is PE:PG:CL (79:19:2) containing acyl chains 15:0, 16:0, 16:1, 18:1, cy17:0. PPM 3.0 determines the energetically optimal spatial position of a protein structure by minimising its free energy of transfer from water to the membrane environment, treating the membrane as a fluid anisotropic solvent.

### Statistical analyses

Statistical analysis of the growth inhibition and immunity-driven recovery assays was performed using GraphPad Prism 9 version 9.5.1. Details regarding specific statistical tests and data interpretation are provided in the corresponding figure legend.

### Structure visualisation tools

Molecular graphics and analyses were performed using UCSF Chimera (Pettersen et al, 2004) and UCSF ChimeraX (Goddard et al, 2018).

## Data availability

The authors declare that the source data supporting the findings of this study are available within the paper and its supplementary information files. The Supplementary Information contains Appendix Figs. S1–S13 and Appendix Tables S1–S3. Atomic coordinates of the Tap3–Tke5 structure have been deposited in the Protein Data Bank (PDB) (accession PDB code id: 9R8G). The cryo-EM map and micrographs are available from the Electron Microscopy Data Bank − EMDB (accession code id: EMD-53820) and the Electron Microscopy Public Image Archive − EMPIAR (accession code id: EMPIAR-13034), respectively. Dataset EV1 is available as a supplementary file and contains the plasmid DNA sequences and the LC-ESI-MS report of the purified Tap3–Tke5 complex. An uncropped and unedited SDS-PAGE image is available online in the Source data.

The source data of this paper are collected in the following database record: biostudies:S-SCDT-10_1038-S44318-025-00689-6.

## Peer review information

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

## Acknowledgements

We acknowledge the Basque Resource for Electron Microscopy (BREM) for access to their cryo-EM facility and their staff for data-acquisition support. The authors would like to thank Diamond Light Source for sample screening and data collection access and support of the cryo-EM facilities at the UK's national Electron Bio-imaging Centre (eBIC) [under proposals EM BI38262], funded by the Wellcome Trust, MRC and BBSRC. We thank Pablo Guerra from the IBMB-CSIC CryoEM Platform for assistance during microscope data acquisition (experiment AV-20240320043). The authors acknowledge funding from Project IU16-014045 (CRYO-TEM) by the Generalitat de Catalunya and by "ERDF A way of making Europe", a European Union initiative. This study was supported by the Spanish Ministry of Science and Innovation (MCIN/AEI/10.13039/501100011033/ FEDER, UE projects PID2021-123000OB-I00, PID2024-159235OB-I00, and CNS2022-135585 to PB, and projects PID2021-127816NB-I00 and PID2024-155225NB-I00 to DAJ) and the Basque Government (IT1745-22 to DAJ). MZZ and JAA acknowledge support from the Basque Government predoctoral program (PRE_2023_1_0100 and PRE_2021_1_0164, respectively). The funders had no role in study design, data collection and analysis, decision to publish, or preparation of the manuscript.

## Author contributions

**Carmen Velázquez**: Data curation; Formal analysis; Validation; Investigation; Visualisation; Methodology; Writing—review and editing. **Maialen Zabala-Zearreta**: Data curation; Formal analysis; Validation; Investigation; Visualisation; Methodology; Writing—review and editing. **Carmen Paredes**: Data curation; Formal analysis; Validation; Investigation; Visualisation; Methodology; Writing—review and editing. **Cristina Civantos**: Data curation; Formal analysis; Validation; Investigation; Visualisation; Methodology; Writing—review and editing. **Jon Altuna-Alvarez**: Data curation; Formal analysis; Validation; Investigation; Visualisation; Methodology; Writing—review and editing. **Patricia Bernal**: Conceptualisation; Resources; Data curation; Formal analysis; Supervision; Funding acquisition; Validation; Investigation; Visualisation; Methodology; Project administration; Writing—review and editing. **David Albesa-Jové**: Conceptualisation; Resources; Data curation; Formal analysis; Supervision; Funding acquisition; Validation; Investigation; Visualisation; Methodology; Writing—original draft; Project administration; Writing—review and editing.

Source data underlying figure panels in this paper may have individual authorship assigned. Where available, figure panel/source data authorship is listed in the following database record: biostudies:S-SCDT-10_1038-S44318-025-00689-6.

## Disclosure and competing interests statement

The authors declare no competing interests.

