## [Peer Review File · The EMBO Journal]

Structural insights into the antibacterial function of the *Pseudomonas putida* effector Tke5

Carmen Velázquez, Maialen Zabala-Zearreta, Carmen Paredes, Cristina Civantos, Jon Altuna-Alvarez, Patricia Bernal, and David Albesa-Jové

Corresponding author(s): David Albesa-Jové (david.albesa@ehu.eus) , Patricia Bernal (pbernal@us.es)

Review Timeline:

Submission Date:	20th Jun 25
Editorial Decision:	15th Aug 25
Revision Received:	12th Nov 25
Editorial Decision:	9th Dec 25
Revision Received:	11th Dec 25
Accepted:	12th Dec 25

Editor: Ioannis Papaioannou

Transaction Report:

Dear Dr. Albesa-Jové,

Thank you again for the submission of your manuscript EMBOJ-2025-121682 for consideration by The EMBO Journal, and for your patience during peer review. Your manuscript has now been seen by four experts in the field, and we have received their detailed, informative, and constructive reports, which you can find below.

I am very pleased to say that, as you will see, the referees indicate interest in your work, explain that your study has been well-designed and performed, the manuscript has been carefully prepared, and the results are novel, relevant, and significant for the field. Referee #4 raises the concern that the importance of Tap-Tke5 interactions in physiologically relevant secretion assays has not been demonstrated, which in their view limits to some extent the impact of the work. In addition, all referees provide lists of suggestions for the improvement of the work and the manuscript. The majority of their comments are rather minor and many of them require only textual revision and clarification to be fully addressed. There are also, however, a few experimental suggestions, which we would strongly encourage you to consider, as they would strengthen your study further and increase its impact on the field. In particular, we would like to draw your attention to the suggestion of referee #3 regarding demonstration that the beta region can interact with membranes, and point 3 of referee #4 regarding the activity of T6-secreted Tke5.

On balance, and given the largely positive referees' comments and recommendations, I would like to invite you to submit a revised version of your manuscript taking the referees' suggestions on board, along with a detailed point-by-point response addressing all referees' comments. Please note that it is The EMBO Journal policy to allow only a single round of major revision, and acceptance of your manuscript will therefore depend on the completeness of your responses in this revised version. Please let me know if you have any questions or comments that you would like to discuss with me.

We generally allow three months as standard revision time (November 14, 2025). As a matter of policy, competing manuscripts published during this period will not negatively impact our assessment of the conceptual advance presented by your study. However, we request that you contact us as soon as possible upon publication of any related work, to discuss how to proceed. Should you foresee a problem in meeting this three-month deadline, please let us know in advance and we will be able to grant an extension.

Thank you for the opportunity to consider your work for publication in The EMBO Journal. I look forward to your revision.

Best regards,

Ioannis

Instructions for preparing your revised manuscript

1. When you are ready to submit the revision, please upload:

- A Word file of the manuscript text (including legends of main Figures, EV Figures and Tables). Please make sure that changes are highlighted (or "tracked") to be clearly visible.

- Individual production-quality figure files (one file per figure). When assembling your figures, please refer to our figure preparation guidelines in order to ensure proper formatting and readability in print as well as on screen:

If the data shown in a figure are obtained from n {less than or equal to} 2, please use scatter plots showing the individual data points.

- i. the name of the statistical test used to generate error bars and P values
- ii. the number (n) of independent experiments (please specify technical or biological replicates) underlying each data point (discussion of statistical methodology can be reported in the Materials and Methods section, but figure legends should contain a basic description of n , P , and the test applied)
- iii. the nature of the bars and error bars (s.d., s.e.m.).

- A point-by-point response to the referees' comments, with a detailed description of the changes made (as a word file). All referees' concerns must be fully addressed and their suggestions taken on board. When preparing your letter of response to the referees' comments, please bear in mind that this will form part of the Review Process File and will therefore be available online to the community. Please note that you have the possibility to opt out of the transparent process at any stage prior to publication by letting the editorial office know (contact@embojournal.org); if you do opt out, the Review Process File link will point to the following statement: "No Review Process File is available with this article, as the authors have chosen not to make the review process public in this case.". For more details on our Transparent Editorial Process, please visit our website: <https://www.embopress.org/page/journal/14602075/authorguide#transparentprocess>

- Expanded View (EV) files (replacing Supplementary Information) that are collapsible/expandable online. A maximum of 5 EV Figures can be typeset. EV Figures should be cited as "Figure EV1, Figure EV2" etc. in the text, and their respective legends should be included in the manuscript file after the legends of regular figures. See detailed instructions regarding Expanded View files here: <https://www.embopress.org/page/journal/14602075/authorguide#expandedview>

- For the figures that you do NOT wish to display as Expanded View figures, they should be bundled together with their legends in a single PDF file called "Appendix", which should start with a short Table of Contents (including page numbers). Appendix figures should be referred to in the main text as: "Appendix Figure S1, Appendix Figure S2" etc. Please see detailed instructions here: <https://www.embopress.org/page/journal/14602075/authorguide#expandedview>

- A complete author checklist, which you can download from our author guidelines (<https://www.embopress.org/page/journal/14602075/authorguide>). Please note that the checklist will also be part of the Review Process File.

2. Please note that no statistics should be calculated and shown in Figures if $n=2$. Please also note that each p value should be reported as an exact value.

3. Before submitting your revision, primary datasets (and computer code, where appropriate) produced in this study need to be deposited in appropriate public databases (see <https://www.embopress.org/page/journal/14602075/authorguide#dataavailability>). The accession numbers, database, and the specific URLs (links) should be listed in a formal "Data availability" section (placed after Methods), following the example below:

"The RNA-seq datasets produced in this study are available in the following database:
Gene Expression Omnibus GSE46843 (<https://www.ncbi.nlm.nih.gov/geo/query/acc.cgi?acc=GSE46843>)"

*** All links should resolve to a page where the data can be accessed. ***

*** Please remember to provide in the Data availability section of your revised manuscript reviewer passwords if the datasets are not yet public. ***

*** The Data Availability Section is restricted to new primary data that are part of this study. In case you have no data that require deposition in a public database, please state so instead of referring to the database: "Our study includes no data deposited in public repositories." under the heading "Data availability". ***

4. The materials and methods need to be described in the manuscript using our structured methods format, which is now required for all research articles. According to this format, the Methods section includes a single "Reagents and Tools Table" - listing key reagents, experimental models, software and relevant equipment including their sources and relevant identifiers - followed by a "Methods and Protocols" section describing the methods. Please download and fill our Reagents and Tools Table template (.docx), which you can find in our author guide:

<https://www.embopress.org/page/journal/14602075/authorguide#structuredmethods>. When submitting your revised manuscript, please do not include the Reagents and Tools Table in the Methods section of the manuscript but instead upload it as a separate file choosing the file type "Reagent Table".

5. Please check that the title and the abstract of the manuscript are brief, yet explicit, even to non-specialists. The length of the title should not exceed 100 characters, and the abstract should be a single paragraph not exceeding 175 words.

6. Please also note our reference format: <https://www.embopress.org/page/journal/14602075/authorguide#referencesformat>.

8. Please remember: digital image enhancement is acceptable practice, as long as it accurately represents the original data and

conforms to community standards. If a figure has been subjected to significant electronic manipulation, this must be noted in the figure legend or in the "Materials and Methods" section. The editors reserve the right to request original versions of figures and the original images that were used to assemble the figure.

9. Our journal encourages inclusion of data citations in the reference list to directly cite datasets that were obtained from public databases. Data citations in the article text are distinct from normal bibliographical citations and should directly link to the database records from which the data can be accessed. In the main text, data citations are formatted as follows: "Data ref: Smith et al, 2001" or "Data ref: NCBI Sequence Read Archive PRJNA342805, 2017". In the Reference list, data citations must be labeled with "[DATASET]". A data reference must provide the database name, accession number/identifiers, and a resolvable link to the landing page from which the data can be accessed at the end of the reference. Further instructions are available at: <https://www.embopress.org/page/journal/14602075/authorguide#referencesformat>.

10. We request authors to consider both actual and perceived competing interests. Please review our policy (<https://www.embopress.org/page/journal/14602075/authorguide#conflictsofinterest>) and update your competing interests statement if necessary. Please name this section 'Disclosure and competing interests statement' and place it after the Acknowledgements section.

11. Please note that all corresponding authors are required to provide an ORCID ID upon submission of a revised manuscript (<https://orcid.org/>). Please find instructions on how to link your ORCID ID to your account in our manuscript tracking system in our Author guidelines (<https://www.embopress.org/page/journal/14602075/authorguide#authorshipguidelines>).

12. We use CRediT to specify the contributions of each author in the journal submission system. CRediT replaces the author contribution section, which should be removed from the manuscript. Please use the free text box to provide more detailed descriptions. See also guide to authors: <https://www.embopress.org/page/journal/14602075/authorguide#authorshipguidelines>.

14. We would also welcome the submission of cover suggestions or motifs to be used by our Graphics Illustrator in designing a cover.

15. Please use the link below to submit your revision:

Referee #1:

In this manuscript, Velasquez and co-workers report the structure of the P. Putida T6SS-delivered toxin protein Tke5, in complex with its cognate T6SS-targeting partner Tap3. This provides important structural insights into this family of pore-inducing toxins, and their interaction with their cognate adapter protein. They also use an elegant structure-function analysis to demonstrate the functional relevance of their interaction to the bactericidal activity of Tke5, providing biological context to the structural work reported in this study.

Overall, this is a well-constructed analysis of this family of toxins, that provides important functional insights into their secretion mechanism, making this a publication well suited for EMBO. I do recommend a number of modifications, to make for a more rigorous analysis, and to facilitate the understanding of this manuscript. Most critically, I find the mechanistic insights into pore formation shown in figure 6 to be a bit of an over-interpretation of the data, considering that there is no direct evidence of the proposed conformational changes for Tke5; and that it is not clear what happens to the the portion of the protein that is not inserted in the membrane according to this model.

Specifically, these elements should be addressed for publication:

- There are some elements of the introduction that are a bit unclear, and referring to figure 1a here would help understand the discussion of the operon; conversely, these are repeated at the beginning of the results section. This could be tightened for clarity.

- Lines 124-127: The authors should describe more clearly the purification procedure, including a description of the tags employed etc...

- Figure 1 is a bit difficult to interpret; I would recommend showing two views of the atomic model, that match the two views of the map.

- A map-to-model FSC should be added to the supplementary material in support of the accuracy of the cryo-EM reconstruction

(easily produced in Phenix). The colours for the local resolution are the opposite of the commonly used colour scheme, with lower resolution typically in blending, not in red as used in Supplementary figure 2.

- Figure 5c and lines 220-222: a schematic representation of the corresponding constructs, with boundaries indicated, would really help understanding the constructs used in this assay.

- Figure 5, and lines 228-233: Why are the scales different between panels C and D? PelB-Tke5 only grows to OD ~ 0.1, Vs ~ 0.4 in C. Also, why is there a decrease in OD after 6h, when co-expressed with Tki5? These need to be either corrected, or explained.

- Figure 6: The authors need to provide pLDDT plot for all AF3 models, else these are not interpretable. I also wonder whether Tke5 could oligomerize in the membrane - AF3 modelling of different oligomeric states, including lipid molecules, could be extremely helpful here.

- Discussion: The mechanism proposed in Figure 6 somehow reminds me of the Botulinum neurotoxin protein, BoNT. Is there any structural similarity with Tke5's a+b domain? Some comparison would be interesting. In addition, it could be interesting to postulate how Tap3 contributes to recruiting Tke5 to the T6SS machinery; this is not addressed here.

Referee #2:

Since the discovery of the MIX domain over ten years ago, the exact role of this domain in effector loading onto the T6SS has been unknown. Here, Velázquez and Zabala-Zearreta and colleagues take an approach of structural biology that complements recent work by the Whitney group (PMID: 39572545) and provides a novel structural and mechanistic understanding of effector loading and folding in the target cell.

In the first part of the manuscript, the authors perform cryo-EM on a protein complex of a T6SS effector and its cognate adaptor protein. This work reveals an unprecedented understanding of the interaction between these proteins via residues of the effector's MIX domain and via an additional loop of the adaptor protein that interacts with the effector outside of the MIX domain. In the second part of the manuscript, the authors complement structural work with an experimental characterization and validation of the effector protein's domains, their role in mediating toxicity, and their interaction with the immunity protein.

To me, these findings present a major break-through in multiple regards. With the MIX domain and adaptor proteins being found in T6SSs across species, the presented findings will apply to many species of Gram-negative bacteria far beyond *P. putida*. The mechanism of effector loading via adaptor proteins is key to the T6SS and I therefore see this work of relevance to the entire T6SS community. The newly discovered protein folds, which do not have structural homology to existing structures in the protein data bank, will further be of interest to the entire community of structural biology.

In my view, the text and the figures are done very well and the data is presented very clearly. The experimental work has been done very thoroughly. I only have minor comments, which I leave up to the authors to address.

Minor comments

- Title, I am wondering, if the title would be stronger if it would become clear what the "mechanism" is about. It reads to me as if there is a mechanism but I do not think I would know whether it is a mechanism of secretion, or mechanism of effector activity,.... Entirely optional, I just thought to point this out.

- L. 55, the authors might consider adding another reference because the 2006 study only reported on the T6SS itself but not yet on its effectors and antibacterial activity.

- L. 71, I am wondering about the reference. Does the referenced study specifically test for the role of the MIX domain in secretion, e.g. by measuring secretion of an effector that has genetically engineered to lack the MIX domain? I don't remember so but might be wrong.

- L. 85, I was a bit surprised about the two references for interspecies and intraspecies competition. They are not wrong, they seem a bit focus on *P. aeruginosa* to me. The authors might want to consider to add references to the first reports on T6SS-mediated inter- and intraspecific killing. To my knowledge, this would be PMID: 20114026 and PMID: 23110230.

- L. 91, I suggest to add a reference to work that also reported on adaptor proteins with a DUF4123 domain in 2015 (PMID: 26150500)

- L. 98, phrasing at the beginning of the sentence, I think it is either "Despite the relevance of..." or "Although the ... proteins are relevant,..."

- L. 100, did the authors consider to mention the recent study by Colautti et al here, which certainly does not cover many aspects of the mechanism but is recent work that provided much more mechanistic insight than there had been before, isn't it? It might be an opportunity for the authors to clearly state again how this work is different/goes beyond the recent study. I see that the authors included a specific reference in l. 180 and also get back to this in the discussion in much detail. I still think it might be worth providing more context to the reader from the start.

- L. 117 "Tap proteins co-evolved with MIX-containing effectors". I guess there is nothing wrong with using the term "co-evolved" here, as this term has been used in the work by Colautti et al, which is correctly referenced. I personally would be a bit careful with the use of the term co-evolution at this moment, I guess this depends a lot on the respective fields though and I am just sharing my thoughts, the authors are free to keep the sentence as is.

- Fig. 5cd, I think there is a typo in the labelling of the y axis. "Growth" should probably read "Growth"
- L. 253ff, I find the idea of a receptor binding domain very intriguing. To make sure I got it right, do the authors think that this RBD will interact with another protein/receptor, that yet needs to be identified? If so, how about indicating such a putative protein/interaction with a question mark into the depiction in figure 5a? Entirely up to the authors. Also for clarity, when the authors write about guiding Tke5 to the "target membrane", what do the authors have in mind when using the term "target membrane"? Is it the inner membrane and not the outer membrane?
- L. 270ff, Fig. 6b, also for clarity. Could you please remind the reader why the MIX domain is not included here or maybe add a sentence to either the figure legend or the text? Is the focus on panel b to purely explain the mechanism and the MIX domain is excluded from the depiction because it is not required for toxicity as shown in the previous figure?

Referee #3:

This is a novel and exciting study that describes the structure of a type VI MIX effector, Tke5, in complex with its adaptor protein, Tap3. The type VI secretion system is a key mechanism that bacteria use to secrete effector proteins that target other bacteria (and in some instances host cells). The T6SS machinery comprises a contractile tail that fires a tube of HCP protein capped by a sharp tip containing bound effector proteins. Understanding how effectors are loaded is a major question in the field. The structure presented provides exciting new information about effector and secretion adaptor complexes. It also shows the structure of a pre-activated form of Tke5, which inhibits target bacteria through the formation of pores in the (cytoplasmic) membrane. The Tap3-bound form of Tke5 masks a proportion of the hydrophobic region that will presumably insert into the target membrane. An intriguing beta-rich region on Tke5 is presumed to bind to target cell membranes, although this is not specifically tested in the manuscript.

It would strengthen the authors' model presented in Figure 6 if they were able to demonstrate that the beta region can interact with membranes.

The experiments that conclude no requirement for the beta region do not strictly test this in the context of T6SS delivery of Tke5, only for delivery of Tke5 to the periplasm by artificially adding the PelB Sec signal sequence. Can the authors test whether the beta region of Tke5 is essential for activity when delivered to target cells by the T6SS?

Minor comments

The authors need to show a global map-to-model fit, ideally with close-up map-to-model fits for the regions of interest discussed in the paper.

What was the FSC cut-off criterion used to estimate the local resolution in Supplementary Figure 2a?

Can the authors show a superposition of the Tke5(608-996) pre-pore state observed in the cryo-EM structure and the predicted pore state? It is difficult to gauge the conformational differences in the two states as presented in Figure 6b.

Can the authors please provide a schematic like in Figure 1a, but with the domains of Tke5 coloured as in Figure 1d?

Referee #4:

This paper reports the cryo EM structure of Tke5 from *Pseudomonas putida* in complex with its DUF4123 adaptor protein. The authors have previously shown that Tke5 is a potent pore forming effector of the *P. putida* type 6 secretion system, and DUF4123 adaptors (now called Tap proteins) are known to be required for effector secretion in *Vibrio cholerae* and *Pseudomonas aeruginosa*. Recent work has described how DUF4123 adaptors link effector toxins to the VgrG spike protein of the T6 secretion apparatus. The current structure shows the interaction between Tap3 and Tke5. Tap3 uses an alpha-helical bundle and an extended loop to interact with the N-terminal MIX domain of Tke5. The authors report that the large central alpha helical portion of Tke5 is likely responsible for the toxin's pore forming activity and provide limited functional data exploring the role of the C-terminal immunoglobulin-like beta-subdomain. In the latter experiment, they find that the beta-subdomain is not required for toxicity and somewhat confusingly conclude that this region may interact with membrane surface to initiate effector penetration. My main criticisms center on the relative lack of Tke5 functional characterization and the plasmid-based overexpression approach to examine Tke5 activity. The structure raises the opportunity to probe the importance of Tap-Tke5 interactions in physiologically relevant secretion assays, but no experiments of this nature are presented. Consequently, the impact of this work is somewhat limited.

Comments

1. The introduction says that Tke5 is a member of the BTH_I2691 protein family that contains an N-terminal MIX domain. The

Uniprot database refers to this domain as the VasX_N domain and a comparison of the AlphaFold model of VasX with that of Tke5 shows many similarities. It's not clear why this effector family is named after BTH_I2691, which has not been characterized experimentally as far as I can see in the literature. It would be less confusing to readers (or at least me) if Tke5 was referred to as a VasX homolog, because this effector has been characterized in a number of *Vibrio* T6SS papers.

2. Results, lines 98 - 100. I don't think this sentence is appropriate given the recent work presented by Colautti et al. (2024). That paper provides significant insight in DUF4123 interactions with VgrG and effectors. Along those lines, does AlphaFold predict how Tap3 interacts with *P. putida* VgrG3?

3. Figures 5c and 5d. This approach to assess toxicity could be flawed. Under physiological conditions, Tke5 would be introduced into the periplasm presumably as a folded structure similar to that resolved by cryo EM. Heterologous expression with an N-terminal signal peptide may not recapitulate the normal membrane insertion mechanism if the central transmembrane helices of Tke5 are inserted directly into the membrane via the lateral gate of SecYEG. The lateral gate is used to insert integral membrane proteins, and therefore this approach could lead the investigators to make erroneous conclusions about the role of the beta Ig-like domain. The paper would be greatly strengthened if the activity of T6 secreted Tke5 was examined.

4. Related to comment 3 above, are Tke5 residues 600-800 sufficient for toxicity if fused to the PelB sequence?

5. I'm not sure that the final section "Proposed Tke5's pore-forming molecular mechanism" is appropriate for the Results. This section is purely speculative and based solely on AlphaFold modeling. As an aside, the model confidence is low (pLLDT <70) to very low (pLLDT <50) for the putative TMD in Fig. 6b. Maybe these passages could be incorporated into the Discussion.

POINT-BY-POINT RESPONSES

Referee #1:

In this manuscript, Velasquez and co-workers report the structure of the *P. Putida* T6SS-delivered toxin protein Tke5, in complex with its cognate T6SS-targeting partner Tap3. This provides important structural insights into this family of pore-inducing toxins, and their interaction with their cognate adapter protein. They also use an elegant structure-function analysis to demonstrate the functional relevance of their interaction to the bactericidal activity of Tke5, providing biological context to the structural work reported in this study.

Overall, this is a well-constructed analysis of this family of toxins, that provides important functional insights into their secretion mechanism, making this a publication well suited for EMBO. I do recommend a number of modifications, to make for a more rigorous analysis, and to facilitate the understanding of this manuscript. Most critically, I find the mechanistic insights into pore formation shown in figure 6 to be a bit of an over-interpretation of the data, considering that there is no direct evidence of the proposed conformational changes for Tke5; and that it is not clear what happens to the portion of the protein that is not inserted in the membrane according to this model.

Specifically, these elements should be addressed for publication:

- There are some elements of the introduction that are a bit unclear, and referring to figure 1a here would help understand the discussion of the operon; conversely, these are repeated at the beginning of the results section. This could be tightened for clarity.

Thanks for the suggestion. We have referenced Fig. 1a in the introduction to discuss the *tke5*-containing operon. Also, we avoid repeating the organization of the operon at the beginning of the results section. The introduction now reads as follows:

Lines 81-84: " The *tke5* gene is found near the 3' end of the K3-T6SS cluster (**Fig. 1a**), where genes coding for a Valine-glycine repeat protein *G* (VgrG3), a Type VI adaptor protein (Tap3), Tke5, its cognate Type six *KT2440* immunity protein (Tki5), and a Type six pair (Tsp5) are also found (Bernal *et al*, 2017; Velázquez *et al*, 2025).

- Lines 124-127: The authors should describe more clearly the purification procedure, including a description of the tags employed etc...

Thanks for the suggestion. We have included a more detailed description of the cloning and purification approaches employed. Now, this section reads as follows:

Lines 130-138: " To demonstrate this, we have solved the 2.8 Å-resolution structure of the Tap3-Tke5 complex using cryo-electron microscopy (cryo-EM) (**Fig. 1, Movie 1, Appendix Figures S1-S3, and Table 1**). To this end, a 6xHis tag was encoded at the 5'

end of *tap3* (*6xhis-tap3*). This construct was inserted at the Multiple Cloning Site 1 (MCS-1) of a pCOLADuetTM-1 plasmid. Then, the unmodified *tke5* gene was inserted at the MCS-2 of pCOLADuetTM-1. The resulting plasmid pCOLADuet-1•*6xhis-tap3-tke5* was transformed into *Escherichia coli* BL21(DE3) for heterologous co-expression of Tap3 and Tke5 (see Methods for details). Following co-expression, we can co-purify the Tap3–Tke5 complex to homogeneity in milligram quantities (Fig. 1b)."

- Figure 1 is a bit difficult to interpret; I would recommend showing two views of the atomic model, that match the two views of the map.

Thanks for the suggestion. We have updated Figure 1 to include two views of the atomic structure, corresponding to the two views of the map. See below:

- A map-to-model FSC should be added to the supplementary material in support of the accuracy of the cryo-EM reconstruction (easily produced in Phenix). The colours for the local resolution are the opposite of the commonly used colour scheme, with lower resolution typically in blending, not in red as used in Supplementary figure 2.

Thanks for the suggestion. We have included a map-to-model FSC in supplementary Fig. 2e, and swapped the colours depicting the local resolution. See below for new Supplementary Figure 2:

- Figure 5c and lines 220-222: a schematic representation of the corresponding constructs, with boundaries indicated, would really help understanding the constructs used in this assay.

Thanks for the suggestion. We have included in Figure 5 a schematic representation of the corresponding constructs. The boundaries of each construct are defined in the text (Lines 223-228) and the corresponding figure legend. See below for the new Figure 5 and text.

Lines 223-228: " To test our hypothesis, we measure the toxicity of Tke5 full-length and deletion constructs when expressed from plasmid pS238D•M in *P. putida* KT2440 (Fig. 5c-d). We engineer the following constructs: Tke5 wild type (MIX- α - β^{1-996}) and a deletion construct lacking the C-terminal β -rich region (MIX- α^{1-863}). Furthermore, we engineer Tke5 constructs containing an N-terminal PelB leader sequence (PelB-MIX- α - β^{1-996}) and deletion constructs PelB-MIX¹⁻²³⁸, PelB- α - $\beta^{239-996}$, PelB- $\alpha^{239-863}$, PelB- $\beta^{864-996}$.

- Figure 5, and lines 228-233: Why are the scales different between panels C and D? PelB-Tke5 only grows to OD ~ 0.1, vs ~ 0.4 in C. Also, why is there a decrease in OD after 6h, when co-expressed with Tki5? These need to be either corrected, or explained.

Thanks for pointing this out to us. Now, panels c and d are on the same scale. Also, we have repeated the toxicity assays and can confirm that the toxicity level of PelB-Tke5 (PelB-MIX- α - β^{1-996}) is similar in panels c and d (OD ~0.1). Furthermore, the new data don't show a decrease in OD after 6h when co-expressed with Tki5.

Previous differences in the OD of panels c and d were likely due to experimental variability that could be caused by the use of different LB batches, 3-mBz stock and/or the switch to an alternate plate reader instrument due to equipment failure. We have repeated the experiment using a single batch of fresh LB, the same 3-mBz stock and the original, repaired plate reader instrument.

- Figure 6: The authors need to provide pLDDT plot for all AF3 models, else these are not interpretable. I also wonder whether Tke5 could oligomerize in the

membrane - AF3 modelling of different oligomeric states, including lipid molecules, could be extremely helpful here.

Thanks for the suggestion. We have included a supplementary figure (**Appendix Figure S13**) showing the AF3 predicted model confidence of Tke5⁶⁰⁸⁻⁹⁹⁶. The AF3 model is colour-coded based on the predicted Local Distance Difference Test (pLDDT) score. Also, we show the Predicted Aligned Error (PAE) plot for Tke5⁶⁰⁸⁻⁹⁹⁶, which shows the expected positional error in angstroms for each residue pair. As indicated by the pLDDT score and the corresponding PAE plot, the model confidence goes from very high (pLDDT > 90) to high (90 > pLDDT > 70) with PAE values between 0-5Å for a large portion of the predicted transmembrane helices (region containing residues 630 to 870) and the immunoglobulin like fold (Ig-like fold; region containing residues 911 to 996). The N-terminal residues that are predicted to form a strand and a short helix (residues 608 to 629) and the loop that protrudes away from the Ig-like fold have very low pLDDT scores (pLDDT < 50). These low values might indicate highly flexible regions. Experimental determination of the Tke5 structure in its membrane-bound conformation is required to validate this prediction. Please see the new supplementary figure below. We have discussed the AF3 predicted model confidence in the **Appendix Figure S13** caption.

Appendix Figure S13. AF3 model confidence for predicted pore state of Tke5⁶⁰⁸⁻⁹⁹⁶. (a) The AF3-predicted structure of Tke5⁶⁰⁸⁻⁹⁹⁶ inserted into a gram-negative inner membrane model, as

suggested by PPM3.0 based on a membrane orientation prediction. The structure is color-coded according to the predicted Local Distance Difference Test (pLDDT) score, a metric indicating the confidence in the predicted atomic positions. The color scale is as follows: blue indicates very high confidence (pLDDT > 90); light blue, high confidence (90 ≥ pLDDT > 70); yellow, moderate confidence (70 ≥ pLDDT > 50); and orange, low confidence (pLDDT ≤ 50). **(b)** Predicted Aligned Error (PAE) plot for Tke5⁶⁰⁸⁻⁹⁹⁶ showing the expected positional error in angstroms for each pair of residues. Dark blue indicates very low predicted error (< 5 Å) and high confidence; yellow indicates moderate confidence with predicted error < 10 Å; orange represents low confidence with predicted error < 15 Å; gray indicates higher error and lower confidence; and white corresponds to very high predicted error (> 30 Å). The reliability of this *in silico* model as evaluated using the per-residue confidence score (pLDDT) and the Predicted Aligned Error (PAE) plot reveals that a significant portion of the model is predicted with high to very high confidence. Specifically, the regions corresponding to the putative transmembrane helices (residues 630 to 870) and the C-terminal immunoglobulin-like (Ig-like) fold (residues 911 to 996) exhibit pLDDT scores largely between 70 and 90, with many exceeding 90. The corresponding PAE plot for these domains shows low expected positional error, with values between 0-5Å, indicating high confidence in the predicted packing and relative orientation of these core structural elements. In contrast, the N-terminal residues (608 to 629), predicted to form a strand and a short helix, along with a loop region protruding from the Ig-like fold, are assigned very low confidence scores (pLDDT < 50). Such low pLDDT values are often indicative of intrinsically disordered or highly flexible regions that do not adopt a single, stable conformation.

Regarding the question of the oligomerization state of Tke5 pore, we thank the reviewer for this insightful question. This is indeed a critical aspect of its mechanism, and we appreciate the suggestion to use AlphaFold modelling to investigate it.

Following the reviewer's suggestion, we performed extensive oligomerization modelling using AlphaFold-Multimer. We tested both the full-length Tke5 protein and the truncated construct Tke5⁶⁰⁸⁻⁹⁹⁶ as potential homodimers and homotrimers. Our analyses revealed that none of these assemblies yielded high-confidence interfaces. The interface-predicted Template Modelling (ipTM) scores, a key metric of interface accuracy, were consistently below 0.4, which is significantly lower than the established threshold for reliable predictions and characteristic of non-interacting proteins. Furthermore, visual inspection of the predicted 3D structures confirmed that they did not form a contiguous transmembrane pore.

These computational results suggest that Tke5 does not form simple, stable dimers or trimers. We posit that this is likely due to two key limitations of current prediction algorithms when applied to this specific biological problem. First, AlphaFold3 predicts static, low-energy structures and cannot model the multi-step, dynamic process of pore formation. Second, and most critically, the algorithm cannot explicitly model the complex, anisotropic environment of a lipid bilayer, which provides the essential thermodynamic driving forces for membrane protein assembly.

Given these inconclusive computational results and the significant experimental challenges associated with definitively determining the stoichiometry of a membrane-embedded pore—which typically requires a combination of specialized techniques such as Blue Native PAGE, chemical cross-linking, and native mass spectrometry—we believe that a full elucidation of the pore architecture represents a substantial future project that builds upon the foundational findings of our current manuscript. To make this clear to the reader, we have now added a statement to the Discussion section that explicitly addresses this point and frames it as an important direction for future research:

Lines 504-509: "A key question for future studies is the precise stoichiometry and architecture of the functional Tke5 pore. While our data clearly demonstrate pore-forming activity leading to selective ion transport, the exact number of subunits required to form the pore remains to be determined. Elucidating the pore's structure will require dedicated biophysical approaches, such as cryo-EM, and represents an exciting avenue for future investigation into the mechanism of the BTH_I2691 toxin family."

- Discussion: The mechanism proposed in Figure 6 somehow reminds me of the Botulinum neurotoxin protein, BoNT. Is there any structural similarity with Tke5's a+b domain? Some comparison would be interesting. In addition, it could be interesting to postulate how Tap3 contributes to recruiting Tke5 to the T6SS machinery; this is not addressed here.

We thank the reviewer for this interesting suggestion. We have performed the requested comparison, and our analysis indicates that Tke5 and BoNT are not homologous. Structural comparison between BoNT and Tke5⁶⁰⁸⁻⁹⁹⁶ indicates low structural and sequence conservation (sequence identity for aligned region is only 13%, overall RMSD is 38 Å). See below for the superposition of BoNT serotype A (PDB ID: 3BTA) with the AF3-predicted Tke5⁶⁰⁸⁻⁹⁹⁶ C-terminal fragment.

Furthermore, they employ fundamentally different mechanisms. BoNT is a canonical A-B toxin targeting eukaryotic neurons. It is internalized via endocytosis, and its light chain translocates to the cytosol, where it functions as

a metalloprotease, cleaving SNARE proteins to block neurotransmitter release. While Tke5 is a T6SS antibacterial effector targeting prokaryotic competitors. It is injected directly into the periplasm, where it inserts into the inner membrane to form ion-selective pores, thereby depolarizing the cell.

Given these profound structural and functional differences, we believe the most mechanistically insightful comparison is not to BoNT, but to other T6SS-delivered pore-forming toxins. As we state in our Discussion section, "Electrophysiological properties and diverse mechanisms of T6SS Pore-Forming Toxins causing bacterial cell depolarization," comparing Tke5 to its functional peers (e.g., Tse4, Tse5) provides a more relevant context for its specific biophysical properties, as these toxins share the same delivery system, cellular compartment, and general mode of action.

Regarding the suggestion to postulate how Tap3 contributes to recruiting Tke5, we agree with the reviewer that this is a fascinating question. However, as this specific interaction was not experimentally explored in our current study, we feel that including a structural prediction or postulation without the relevant validation would be too speculative. We have therefore focused our manuscript on the novel Tap3-Tke5 interactions and the pore-forming activity of Tke5, for which we provide direct experimental and structural evidence.

Referee #2:

Since the discovery of the MIX domain over ten years ago, the exact role of this domain in effector loading onto the T6SS has been unknown. Here, Velázquez and Zabala-Zearreta and colleagues take an approach of structural biology that complements recent work by the Whitney group (PMID: 39572545) and provides a novel structural and mechanistic understanding of effector loading and folding in the target cell.

In the first part of the manuscript, the authors perform cryo-EM on a protein complex of a T6SS effector and its cognate adaptor protein. This work reveals an unprecedented understanding of the interaction between these proteins via residues of the effector's MIX domain and via an additional loop of the adaptor protein that interacts with the effector outside of the MIX domain. In the second part of the manuscript, the authors complement structural work with an experimental characterization and validation of the effector protein's domains, their role in mediating toxicity, and their interaction with the immunity protein.

To me, these findings present a major break-through in multiple regards. With the MIX domain and adaptor proteins being found in T6SSs across species, the presented findings will apply to many species of Gram-negative bacteria far beyond *P. putida*. The mechanism of effector loading via adaptor proteins is key to the T6SS and I therefore see this work of relevance to the entire T6SS community. The newly discovered protein folds, which do not have structural homology to existing structures in the protein data bank, will further be of interest to the entire community of structural biology.

In my view, the text and the figures are done very well and the data is presented very clearly. The experimental work has been done very thoroughly. I only have minor comments, which I leave up to the authors to address.

Minor comments

- Title, I am wondering, if the title would be stronger if it would become clear what the "mechanism" is about. It reads to me as if there is a mechanism but I do not think I would know whether it is a mechanism of secretion, or mechanism of effector activity,.... Entirely optional, I just thought to point this out.

Thanks for the suggestion. We have modified the title to better convey the impact of the results, while limiting the length to ~100 characters following editorial policy:

"Structural basis for secretion and activity of a widespread Type VI secretion system pore-forming toxin family"

- L. 55, the authors might consider adding another reference because the 2006 study only reported on the T6SS itself but not yet on its effectors and antibacterial activity.

Thanks for the suggestion. We have referenced one more seminal paper that demonstrates the antibacterial activity of T6SS-delivered effectors:

Line 52: (Mougous *et al*, 2006; Russell *et al*, 2011)

- L. 71, I am wondering about the reference. Does the referenced study specifically test for the role of the MIX domain in secretion, e.g. by measuring secretion of an effector that has genetically engineered to lack the MIX domain? I don't remember so but might be wrong.

Thanks for pointing this out to us. The referenced study was the first to identify that several T6SS effectors contain the MIX motif; however, as you indicate, it did not evaluate the role of the MIX motif in secretion. So, we have included a second reference that experimentally demonstrates that a DNase Type VI secretion system effector requires its MIX domain for secretion (<https://doi.org/10.1128/spectrum.02465-22>).

Lines 70: (Salomon *et al*, 2014; Fridman *et al*, 2022)

- L. 85, I was a bit surprised about the two references for interspecies and intraspecies competition. They are not wrong, they seem a bit focus on *P. aeruginosa* to me. The authors might want to consider to add references to the first reports on T6SS-mediated inter- and intraspecific killing. To my knowledge, this would be PMID: 20114026 and PMID: 23110230.

Thanks for the suggestion. We have included both references in the manuscript:

Lines 91-94: "Antiprobkaryotic effector proteins delivered by the T6SS are often paired with cognate immunity proteins that protect the producing cell from self-intoxication, as well as interspecies (Basler *et al*, 2013; Hood *et al*, 2010) and intraspecies (George *et al*, 2024; Unterweger *et al*, 2012) competition."

- L. 91, I suggest to add a reference to work that also reported on adaptor proteins with a DUF4123 domain in 2015 (PMID: 26150500)

Thanks for the suggestion. We have included both references in the manuscript:

Lines 91-94: "Antiprobkaryotic effector proteins delivered by the T6SS are often paired with cognate immunity proteins that protect the producing cell from self-intoxication, as well as interspecies (Basler *et al*, 2013; Hood *et al*, 2010) and intraspecies (George *et al*, 2024; Unterweger *et al*, 2012) competition."

- L. 98, phrasing at the beginning of the sentence, I think it is either "Despite the relevance of..." or "Although the ... proteins are relevant,..."

Thanks for the correction. We have rewritten this paragraph in order to introduce previous work on the *Pseudomonas aeruginosa* Tap6.

- L. 100, did the authors consider to mention the recent study by Colautti *et al* here, which certainly does not cover many aspects of the mechanism but is recent work that provided much more mechanistic insight than there had been before, isn't it? It might be an opportunity for the authors to clearly state again how this work is different/goes beyond the recent study. I see that the authors included a specific reference in l. 180 and also get back to this in the discussion in much detail. I still think it might be worth providing more context to the reader from the start.

Thanks for the suggestion. We have introduced the recent findings by Colautti *et al* to provide context to the reader from the start:

Line 106-112: "Previous work on a Tap from *Pseudomonas aeruginosa* (Tap6) demonstrated that it directly binds its cognate effector, Ptx2. Experiments confirmed its essential role in Ptx2 export and suggested that Tap6's variable C-terminal lobe mediates effector recognition (Colautti *et al*, 2024). Nonetheless, despite the importance of T6SS adaptor proteins, the molecular mechanisms they use to bind to their cognate effectors have not been experimentally validated. To bridge this knowledge gap, we have determined the 2.8 Å cryo-EM structure of Tke5 in complex with its cognate adaptor Tap3.

- L. 117 "Tap proteins co-evolved with MIX-containing effectors". I guess there is nothing wrong with using the term "co-evolved" here, as this term has been used in the work by Colautti *et al*, which is correctly referenced. I personally would be a bit careful with the use of the term co-evolution at this moment, I guess this depends a lot on the respective fields though and I am just sharing my thoughts, the authors are free to keep the sentence as is.

Thanks for the suggestion. We have deleted the sentence "Tap proteins co-evolved with MIX-containing effectors".

- Fig. 5cd, I think there is a typo in the labelling of the y axis. "Growth" should probably read "Growth"

Thanks for the correction.

- L. 253ff, I find the idea of a receptor binding domain very intriguing. To make sure I got it right, do the authors think that this RBD will interact with another protein/receptor, that yet needs to be identified? If so, how about indicating such a putative protein/interaction with a question mark into the depiction in figure 5a? Entirely up to the authors.

Based on computational predictions, we initially thought the RBD could bind the target inner membrane. Still, we couldn't exclude the possibility that it can also recognise an unknown membrane protein receptor. We have now obtained experimental data supporting this initial prediction. In view of the new data, we have included a new Results section: **"The C-terminal β -rich region of Tke5 binds to model membranes *in vitro*"**

In this section, we present the results of a fluorescence co-localization assay using a GFP-RBD fusion protein and Giant Unilamellar Vesicles (GUVs). Our results show that the GFP-RBD protein, but not a GFP-only control, directly co-localizes with the GUVs, providing experimental support for the capacity of the RBD to bind membranes. Furthermore, this interaction occurred with GUVs composed of both complex *E. coli* polar lipids and simple DOPC, suggesting the RBD functions as a membrane-tethering module that does not require specific lipid head groups.

These new results are also presented in Figure 6:

Fig. 6 | The C-terminal β -rich region of Tke5 binds to model membranes *in vitro*. (a) Model of Tke5 interaction with the inner membrane of a target bacterium. Following

delivery into the periplasmic space via the T6SS, Tke5 is hypothesised to initially interact with the inner membrane through its C-terminal β -rich receptor-binding domain (RBD, residues 864-996, shown in red). This initial binding, predicted by PPM 3.0, occurs via the distal end of the RBD, including a modelled disordered loop (residues 886-909). The "FR" label points to a region in Tke5 (residues 319-365, shown in black) that, in the homolog Ptx2, was proposed to function as a membrane sensor. **(b)** Binding of GFP-RBD to Giant Unilamellar Vesicles (GUVs). GUVs composed of *E. coli* polar lipid extract or pure DOPC were labelled with rhodamine-DHPE (red channel) and incubated with either GFP-RBD or GFP alone (green channel). Images show the GFP, rhodamine, and merged fluorescence signals. GFP-RBD colocalises with the vesicle membrane, whereas GFP signal shows no detectable membrane binding. Source data are available online for this figure.

Also for clarity, when the authors write about guiding Tke5 to the "target membrane", what do the authors have in mind when using the term "target membrane"? Is it the inner membrane and not the outer membrane?

We have changed "target membrane" for "target inner membrane" to make it clearer:

Line 273-275: "This membrane-sensing role of the RBD could be relevant for guiding Tke5 to its target inner membrane and potentially orienting it for subsequent insertion."

- L. 270ff, Fig. 6b, also for clarity. Could you please remind the reader why the MIX domain is not included here or maybe add a sentence to either the figure legend or the text? Is the focus on panel b to purely explain the mechanism and the MIX domain is excluded from the depiction because it is not required for toxicity as shown in the previous figure?

Thanks for the suggestion. We have revised Fig. 6b (**now Fig. 7**) legend to make it clear why we are only depicting the C-terminal half of Tke5:

Lines 1058-1059: " The Tke5 N-terminal region (residues 1-607), which includes the MIX domain, is not shown for clarity."

Referee #3:

This is a novel and exciting study that describes the structure of a type VI MIX effector, Tke5, in complex with its adaptor protein, Tap3. The type VI secretion system is a key mechanism that bacteria use to secrete effector proteins that target other bacteria (and in some instances host cells). The T6SS machinery comprises a contractile tail that fires a tube of HCP protein capped by a sharp tip containing bound effector proteins. Understanding how effectors are loaded is a major question in the field. The structure presented provides exciting new information about effector and secretion adaptor complexes. It also shows the structure of a pre-activated form of Tke5, which inhibits target bacteria through the formation of pores in the (cytoplasmic) membrane. The Tap3-bound form of

Tke5 masks a proportion of the hydrophobic region that will presumably insert into the target membrane. An intriguing beta-rich region on Tke5 is presumed to bind to target cell membranes, although this is not specifically tested in the manuscript.

It would strengthen the authors' model presented in Figure 6 if they were able to demonstrate that the beta region can interact with membranes.

We thank the reviewer for the suggestion. We have now obtained experimental data supporting the idea that the beta-rich region on Tke5 might be able to bind to target cytoplasmic membranes. In view of the new data, we have included a new Results section: **"The C-terminal β -rich region of Tke5 binds to model membranes *in vitro*"**

In this section, we present the results of a fluorescence co-localization assay using a GFP-RBD fusion protein and Giant Unilamellar Vesicles (GUVs). Our results show that the GFP-RBD protein, but not a GFP-only control, directly co-localizes with the GUVs, providing experimental support for the capacity of the RBD to bind membranes. Furthermore, this interaction occurred with GUVs composed of both complex *E. coli* polar lipids and simple DOPC, suggesting the RBD functions as a membrane-tethering module that does not require specific lipid head groups.

These new results are also presented in Figure 6:

Fig. 6 | The C-terminal β -rich region of Tke5 binds to model membranes *in vitro*. (a) Model of Tke5 interaction with the inner membrane of a target bacterium. Following delivery into the periplasmic space via the T6SS, Tke5 is hypothesised to initially interact with the inner membrane through its C-terminal β -rich receptor-binding domain (RBD, residues 864-996, shown in red). This initial binding, predicted by PPM 3.0, occurs via the distal end of the RBD, including a modelled disordered loop (residues 886-909). The "FR" label points to a region in Tke5 (residues 319-365 shown in black) that was proposed in the homolog Ptx2 to function as a membrane sensor. (b) Binding of GFP-RBD to Giant Unilamellar Vesicles (GUVs). GUVs composed of *E. coli*

polar lipid extract or pure DOPC were labelled with rhodamine-DHPE (red channel) and incubated with either GFP-RBD or GFP alone (green channel). Images show the GFP, rhodamine, and merged fluorescence signals. GFP-RBD colocalises with the vesicle membrane, whereas GFP signal shows no detectable membrane binding. Source data are available online for this figure.

The experiments that conclude no requirement for the beta region do not strictly test this in the context of T6SS delivery of Tke5, only for delivery of Tke5 to the periplasm by artificially adding a the PelB Sec signal sequence. Can the authors test whether the beta region of Tke5 is essential for activity when delivered to target cells by the T6SS?

Thanks for the comment. We agree with the reviewer that testing Tke5 activity in the context of a T6SS delivery assay would be ideal. Unfortunately, the activity of T6-secreted Tke5 cannot be evaluated because the K3-T6SS associated with Tke5 is not active under laboratory conditions. But we can simulate the native-like membrane insertion mechanism of Tke5 *in vivo* by expressing wild-type Tke5 in the bacterial cytoplasm without including the PelB signal sequence.

Expression of Tke5 wild type in the cytoplasm of bacteria is also toxic, although to a lesser extent than when Tke5 is fused to an N-terminal signal peptide. Furthermore, this toxicity can be neutralised if Tke5 is co-expressed with its cognate immunity protein Tki5. Also, a deletion construct lacking the C-terminal beta-rich region (MIX- α^{1-863}) has the same level of toxicity as wild type Tke5 (MIX- α - β^{1-996}), and bacteria co-expressing the immunity protein are also protected from its toxicity. Altogether, our data support the idea that the alpha-helical region alone is sufficient to kill intoxicated bacteria, and Tki5 neutralises its effect rather than interfering with other domains.

The new toxicity assays containing the results with Tke5 wild type and deletion construct MIX- α^{1-863} have been included in the Fig. 5d and the Results section: **"The α -helical region of Tke5 is essential for toxicity and immunity-mediated neutralisation."** See below for details:

Lines 223-260: " To test our hypothesis, we measure the toxicity of Tke5 full-length and deletion constructs when expressed from plasmid pS238D•M in *P. putida* KT2440 (Fig. 5c-d). We engineer the following constructs: Tke5 wild type (MIX- α - β^{1-996}) and a deletion construct lacking the C-terminal β -rich region (MIX- α^{1-863}). Furthermore, we engineer Tke5 constructs containing an N-terminal PelB leader sequence (PelB-MIX- α - β^{1-996}) and deletion constructs PelB-MIX $^{1-238}$, PelB- α - $\beta^{239-996}$, PelB- $\alpha^{239-863}$, PelB- $\beta^{864-996}$.

Previously established, the expression of Tke5 is toxic to *P. putida* KT2440, an effect that is neutralised by the co-expression of its cognate immunity protein, Tki5. The toxin's potency is significantly enhanced by the addition of a PelB leader sequence, which targets the protein for secretion into the periplasm (Velázquez *et al*, 2025). This increased toxicity is the expected outcome, as it aligns with previous *in vitro* electrophysiological data showing that Tke5 senses the directionality of the membrane

potential and is expected to preferentially insert into the inner membrane from the periplasmic side (Velázquez *et al*, 2025).

To pinpoint the toxic component of Tke5, we have systematically analysed the series of deletion constructs generated. Our findings show that the C-terminal β -rich region is not essential for toxicity, as its deletion (PelB- $\alpha^{239-863}$ or MIX- α^{1-863}) does not eliminate the toxic effect (Fig. 5c-d). Furthermore, constructs lacking the N-terminal MIX domain (PelB- α - $\beta^{239-996}$ or PelB- $\alpha^{239-863}$) remain as toxic as the full-length protein PelB-MIX- α - β^{1-996} , while the MIX domain (PelB-MIX $^{1-238}$) or the C-terminal β -rich region (PelB- $\beta^{864-996}$) alone are non-toxic (Fig. 5c). Crucially, the minimal construct containing only the α -helical region (PelB- $\alpha^{239-863}$) is sufficient to induce cell death, demonstrating that this domain is the core toxic module of the effector (Fig. 5c).

To validate that this observed toxicity is the result of Tke5's specific activity, we have co-expressed the toxic constructs with the cognate immunity protein (Fig. 5d). The presence of Tki5 fully protected the cells from the effects of full length Tke5 (MIX- α - β^{1-996} , PelB-MIX- α - β^{1-996}), and deletion constructs containing the α -helical region (MIX- α^{1-863} , PelB- $\alpha^{239-863}$). Taken together, these results confirm that the immunity protein acts by directly neutralising the α -helical region.

In summary, while our previous work identified Tke5 as a bactericidal toxin that causes cell depolarisation, our current findings establish that this activity is mediated entirely by its α -helical region. This region alone is sufficient to kill intoxicated bacteria, and the ability of Tki5 to neutralise its effect confirms that the immunity protein directly counteracts the effector's toxic activity rather than interfering with other domains.

Fig. 5 | The α -helical region of Tke5 mediates toxicity and is neutralised by its cognate immunity protein Tki5. (a) Domain architecture of Tke5 showing the N-terminal MIX

domain (residues 17–238, blue), α -helical pore-forming region (residues 239–863, grey and green), and C-terminal β -rich region (residues 864–996, green). Predicted transmembrane helices (green) are mapped onto the α -helical region, which corresponds to residues 622-644 (FLGLLTSGGGGGLNAGMLWFNIL), 663-685 (LGFASSVFGVIGAAAATLVSV-RA), 695-717 (ISTAPGMAFGNGIINFLTSNLFA), 744-766 (TALGSILVL) and 781-803 (FAGWTAAGIALIGATLIGGGFL). **(b)** Two 180° rotated views of the β -rich region that presents an immunoglobulin-like sandwich fold located at the C-terminal of Tke5. **(c)** Toxicity assays of PelB-containing Tke5 and truncation constructs in *P. putida*. Full-length Tke5 (PelB·MIX- α - β ¹⁻⁹⁹⁶), the α + β domain (PelB· α - β ²³⁹⁻⁹⁹⁶), and the α -helical region alone (PelB· α ²³⁹⁻⁸⁶³) show comparable toxicity. The MIX domain (PelB·MIX¹⁻²³⁸) and β -rich region (PelB· β ⁸⁶⁴⁻⁹⁹⁶) are non-toxic, with bacterial growth comparable to that of bacteria transformed with the empty vector (EV). **(d)** Neutralisation of Tke5 toxicity by its cognate immunity protein Tki5. Co-expression of Tki5 rescues cell growth inhibited by PelB·MIX- α - β ¹⁻⁹⁹⁶, PelB· α ²³⁹⁻⁸⁶³, MIX- α - β ¹⁻⁹⁹⁶ and MIX- α ¹⁻⁸⁶³, confirming the specificity of Tki5 for the α -helical toxic domain. Expression of constructs MIX- α - β ¹⁻⁹⁹⁶ or MIX- α ¹⁻⁸⁶³ that lack the PelB leader sequence are also toxic, although to a less extent than PelB-containing constructs. A schematic representation of the corresponding constructs is shown next to corresponding plot. Plotted data are average of 3 or 6 independent biological replicates (n = 3, 6). Error bars correspond to standard deviations (SD). Source data are available online for this figure.

Minor comments

The authors need to show a global map-to-model fit, ideally with close-up map-to-model fits for the regions of interest discussed in the paper.

Thanks for the suggestion. We have included a new supplementary figure (Appendix Figure S3) showing a global map-to-model fit and close-up map-to-model fits for the regions of interest discussed in the paper. See below for new Appendix Figure S3:

a)

b)

c)

Appendix Figure S3. Tap3-Tke5 Map-to-model fits.

Map-to-model fit of a) side and top/bottom view of Tap3–Tke5 complex, b) zoomed views of interaction site residues between Tap3 N- and C-terminal and Tke5, and c) detailed view of the interface between MIX domain and $\alpha\beta$ domain of Tke5. All views were generated in ChimeraX.

What was the FSC cut-off criterion used to estimate the local resolution in Supplementary Figure 2a?

The resolution estimation was based on an FSC threshold of 0.5. This information has been included in the Suppl. Fig. 2a legend.

Can the authors show a superposition of the Tke5(608-996) pre-pore state observed in the cryo-EM structure and the predicted pore state? It is difficult to gauge the conformational differences in the two states as presented in Figure 6b.

Thanks for the suggestion. We modified Fig. 6b (**now Fig. 7b**) to make it easier to gauge the conformational differences in the C-terminal Tke5⁶⁰⁸⁻⁹⁹⁶ pre-pore and pore states. In particular, the residues predicted to assemble the transmembrane domain are shown with a rainbow colour scheme. Also, we have included the atomic coordinates and JSON file of the AF3 prediction for Tke5⁶⁰⁸⁻⁹⁹⁶ pore state in the "source data" that accompanies this manuscript. See below for the new figure.

Fig. 6 | The C-terminal β -rich region of Tke5 binds to model membranes *in vitro*. (a) Model of Tke5 interaction with the inner membrane of a target bacterium. Following delivery into the periplasmic space via the T6SS, Tke5 is hypothesised to initially interact with the inner membrane through its C-terminal β -rich receptor-binding domain (RBD, residues 864-996, shown in red). This initial binding, predicted by PPM 3.0, occurs via the distal end of the RBD, including a modelled disordered loop

(residues 886-909). The "FR" label points to a region in Tke5 (residues 319-365 shown in black) that was proposed in the homolog Ptx2 to function as a membrane sensor. **(b)** Binding of GFP-RBD to Giant Unilamellar Vesicles (GUVs). GUVs composed of *E. coli* polar lipid extract or pure DOPC were labelled with rhodamine-DHPE (red channel) and incubated with either GFP-RBD or GFP alone (green channel). Images show the GFP, rhodamine, and merged fluorescence signals. GFP-RBD colocalises with the vesicle membrane, whereas GFP signal shows no detectable membrane binding. Source data are available online for this figure.

Can the authors please provide a schematic like in Figure 1a, but with the domains of Tke5 coloured as in Figure 1d?

Thanks for the suggestion. We have modified Fig. 1a to show the domains of Tke5 coloured as in Fig. 1d. Also, we have included the names used to define each domain and region of Tke5 in Fig. 1d. See below for the updated figure:

Referee #4:

This paper reports the cryo EM structure of Tke5 from *Pseudomonas putatida* in complex with its DUF4123 adaptor protein. The authors have previously shown that Tke5 is a potent pore forming effector of the *P. putida* type 6 secretion system, and DUF4123 adaptors (now called Tap proteins) are known to be required for effector secretion in *Vibrio cholerae* and *Pseudomonas aeruginosa*. Recent work has described how DUF4123 adaptors link effector toxins to the VgrG spike protein of the T6 secretion apparatus. The current structure shows the interaction between Tap3 and Tke5. Tap3 uses an alpha-helical bundle and an extended loop to interact with the N-terminal MIX domain of Tke5. The authors report that the large central alpha helical portion of Tke5 is likely responsible for the toxin's pore forming activity and provide limited functional data exploring the role of the C-terminal immunoglobulin-like beta-subdomain. In the latter experiment, they find that the beta-subdomain is not required for toxicity and somewhat confusingly conclude that this region may interact with membrane surface to initiate effector penetration. My main criticisms center on the relative lack of Tke5 functional characterization and the plasmid-based overexpression approach to examine Tke5 activity. The structure raises the opportunity to probe the importance of Tap-Tke5 interactions in physiologically relevant secretion assays, but no experiments of this nature are presented. Consequently, the impact of this work is somewhat limited.

Comments

1. The introduction says that Tke5 is a member of the BTH_I2691 protein family that contains an N-terminal MIX domain. The Uniprot database refers to this domain as the VasX_N domain and a comparison of the AlphaFold model of VasX with that of Tke5 shows many similarities. It's not clear why this effector family is named after BTH_I2691, which has not been characterized experimentally as far as I can see in the literature. It would be less confusing to readers (or at least me) if Tke5 was referred to as a VasX homolog, because this effector has been characterized in a number of *Vibrio* T6SS papers.

We thank the reviewer for this insightful comment. In our manuscript, we refer to Tke5 as “a member of the T6SS effector BTH_I2691 protein family (InterPro accession NF041559)” based on the family designation provided by the European Bioinformatics Institute (EBI). This protein family includes both Tke5 and VasX, among other homologous T6SS effectors. Although the family is named after the *Burkholderia thailandensis* protein BTH_I2691—which, as the reviewer correctly notes, has not been experimentally characterized—this nomenclature is derived from InterPro's classification system, not from functional precedence in the literature.

To avoid confusion and to reflect the reviewer's helpful suggestion, we have revised the manuscript to clarify that VasX is a well-characterized and representative member of the T6SS effector BTH_I2691 protein family. This additional context will help readers better understand the relationship between Tke5 and VasX and the rationale behind the protein family naming convention. Also, we have extended the introduction section to include all the VasX

homologues that have been experimentally studied, these include: VasX, Ptx2 and Bte2. See below for updated introduction:

Lines 58-79: "Among the effectors encoded within the K3-T6SS cluster of *P. putida* KT2440, a recent study identified a Type six KT2440 effector 5 (Tke5) as a potent antibacterial toxin effective against ten of the most resilient plant pathogens (Velázquez *et al*, 2025). Notably, this work also elucidated Tke5's molecular mechanism of action, showing that it functions as a novel pore-forming toxin (PFT) that kills target cells through selective ion transport. Instead of causing extensive membrane disruption, Tke5 induces membrane depolarisation by perturbing ion homeostasis, ultimately leading to bacterial cell death while preserving overall membrane integrity (Velázquez *et al*, 2025). This work represented the first biophysical dissection for a member of the T6SS effector BTH_I2691 protein family (InterPro accession NF041559), also commonly referred to as the VasX family. Members of this family are large, multi-domain pore-forming toxins harbouring an N-terminal marker for type six secretion system effectors (MIX) motif that was shown to be necessary for T6SS-dependent secretion (Salomon *et al*, 2014; Fridman *et al*, 2022). Those family members that have been experimentally studied to date illustrate a remarkable evolutionary diversification in their membrane-targeting strategies. These include the archetypal *Vibrio cholerae* effector VasX, a toxin capable of killing both eukaryotic cells, such as the amoeba *Dictyostelium discoideum*, and prokaryotic competitors by engaging membranes via direct lipid-binding (Miyata *et al*, 2011, 2013); the structural prototype *Pseudomonas aeruginosa* Ptx2 (Colautti *et al*, 2025); and the highly divergent *Bacteroides fragilis* effector Bte2, a gut-adapted specialist that has uniquely co-opted the Rnf electron transport complex in target cells as a receptor to mediate intoxication (Ratner *et al*, 2025)."

2. Results, lines 98 - 100. I don't think this sentence is appropriate given the recent work presented by Colautti et al. (2024). That paper provides significant insight in DUF4123 interactions with VgrG and effectors. Along those lines, does Alphafold predict how Tap3 interacts with *P. putida* VgrG3?

We appreciate the reviewer's insightful feedback and for highlighting the recent, important work by Colautti et al. (2024). We agree that our original sentence did not adequately reflect the significant contributions of this study.

Accordingly, we have rewritten this section to properly acknowledge their findings. This revision also serves to better contextualize the specific contribution of our work, which provides the first structural insights into an adaptor-effector complex.

Regarding the excellent question about the predicted interaction between Tap3 and *P. putida* VgrG3, we agree this is a fascinating area for future investigation. However, as this specific interaction was not experimentally explored in our study, we feel that including a structural prediction without the relevant validation would be speculative. We have therefore focused our discussion on the Tap3-Tke5 interactions for which we provide direct experimental evidence, and on the pore-forming activity of Tke5.

There it is the corresponding rewritten section in the introduction of the manuscript:

Lines 106-119: "Previous work on a Tap from *Pseudomonas aeruginosa* (Tap6) demonstrated that it directly binds its cognate effector, Ptx2. Experiments confirmed its essential role in Ptx2 export and suggested that Tap6's variable C-terminal lobe mediates effector recognition (Colautti *et al*, 2024). Nonetheless, despite the importance of T6SS adaptor proteins, the molecular mechanisms they use to bind to their cognate effectors have not been experimentally validated. To bridge this knowledge gap, we have determined the 2.8 Å cryo-EM structure of Tke5 in complex with its cognate adaptor Tap3. Our structure uncovers how Tap3 recognises the MIX domain of Tke5. Furthermore, we define the domain architecture of Tke5 and demonstrate that only an α -helical region located towards its C-terminal end is sufficient to kill intoxicated bacteria. In contrast, a β -rich region at the C-terminus is likely to enhance target membrane specificity. Together, our results provide a comprehensive framework for understanding how similar MIX-containing BTH_I2691 effectors might be recognised by their cognate Tap proteins, and we dissect their structural organisation and mechanism of action."

Also, we would like to point out that during the revision of this manuscript the cryo-EM structure of Ptx2 was released. Therefore, we have mention it in the introduction and compare it with Tke5 in the discussion. See below for corresponding text and figures included in the manuscript:

Lines 337-361: The challenges in computationally modelling the Tap6-Ptx2 interaction were mirrored by experimental hurdles. In a recent study, Colautti and colleagues attempted to determine the structure of the Tap6–Ptx2 complex using cryo-EM. These efforts were ultimately unsuccessful in yielding a 3D reconstruction, a difficulty the authors attributed to the low number of particles forming the complex and significant flexibility at the interaction site (Colautti *et al*, 2025). Nevertheless, their analysis of 2D class averages did reveal additional density near the N-terminal MIX domain of Ptx2, consistent with Tap6 binding at this location. These prior challenges highlight the significance of our Tap3-Tke5 structure, which successfully overcomes these obstacles to provide the first atomic-level visualisation of an adaptor-effector complex from this widespread family.

The recent cryo-EM structure of Ptx2 provides the first opportunity for a direct structural comparison between two members of the BTH_I2691 family (Colautti *et al*, 2025). A structural alignment of our Tke5 structure with that of Ptx2 reveals that while the overall domain organisation is conserved, the proteins are remarkably divergent (Fig. 7a). Both effectors consist of an N-terminal MIX domain, a large central α -helical region that contains a predicted transmembrane domain (TMD), and a C-terminal β -rich region with an immunoglobulin-like fold. Despite this shared architecture, the overall structures differ significantly, with a root-mean-square deviation (RMSD) across aligned C α atom pairs of 25 Å. This divergence is evident across all domains; when structurally aligned, the MIX domains, the α -helical regions, and the C-terminal β -rich regions show marked differences, with RMSD values of 6.2 Å, 22.5 Å, and 21.9 Å, respectively (Appendix Figure S12). In Ptx2, the large α -helical scaffold domain forms a

"claw-like" structure that completely encloses its TMD, a feature that is broadly consistent with the architecture of Tke5's α -helical region.

New Figure 7:

New Appendix Figure S12:

3. Figures 5c and 5d. This approach to assess toxicity could be flawed. Under physiological conditions, Tke5 would be introduced into the periplasm presumably as a folded structure similar to that resolved by cryo-EM. Heterologous expression with an N-terminal signal peptide may not recapitulate the normal membrane insertion mechanism if the central transmembrane helices of Tke5 are inserted directly into the membrane via the lateral gate of SecYEG. The lateral gate is used to insert integral membrane proteins, and therefore this approach could lead the investigators to make erroneous conclusions about the role of the beta Ig-like domain. The paper would be greatly strengthened if the activity of T6 secreted Tke5 was examined.

Thanks for the comment. We agree that heterologous expression with an N-terminal signal peptide may not recapitulate the normal membrane insertion mechanism because the central transmembrane helices of Tke5 are inserted directly into the membrane via the lateral gate of SecYEG.

Unfortunately, the activity of T6 secreted Tke5 cannot be evaluated because the K3-T6SS that is associated with Tke5 is not active under laboratory conditions.

But we can simulate the native-like membrane insertion mechanism of Tke5 *in vivo* by expressing wild-type Tke5 in the bacterial cytoplasm.

Expression of Tke5 wild type in the cytoplasm of bacteria is also toxic, although to a lesser extent than when Tke5 is fused to an N-terminal signal peptide. Furthermore, this toxicity can be neutralised if Tke5 is co-expressed with its cognate immunity protein Tki5. Also, a deletion construct lacking the C-terminal beta-rich region (MIX- α^{1-863}) has the same level of toxicity as wild-type Tke5 (MIX- α - β^{1-996}), and bacteria co-expressing the immunity protein are also protected from its toxicity. Altogether, our data support the idea that the alpha-helical region alone is sufficient to kill intoxicated bacteria, and Tki5 neutralises its effect rather than interfering with other domains.

The new toxicity assays containing the results with Tke5 wild type and deletion construct MIX- α^{1-863} have been included in the Fig. 5d and the Results section: **"The α -helical region of Tke5 is essential for toxicity and immunity-mediated neutralisation."** See below for details:

Lines 223-260: "To test our hypothesis, we measure the toxicity of Tke5 full-length and deletion constructs when expressed from plasmid pS238D•M in *P. putida* KT2440 (Fig. 5c-d). We engineer the following constructs: Tke5 wild type (MIX- α - β^{1-996}) and a deletion construct lacking the C-terminal β -rich region (MIX- α^{1-863}). Furthermore, we engineer Tke5 constructs containing an N-terminal PelB leader sequence (PelB-MIX- α - β^{1-996}) and deletion constructs PelB-MIX $^{1-238}$, PelB- α - $\beta^{239-996}$, PelB- $\alpha^{239-863}$, PelB- $\beta^{864-996}$.

Previously established, the expression of Tke5 is toxic to *P. putida* KT2440, an effect that is neutralised by the co-expression of its cognate immunity protein, Tki5. The toxin's potency is significantly enhanced by the addition of a PelB leader sequence, which targets the protein for secretion into the periplasm (Velázquez *et al*, 2025). This increased toxicity is the expected outcome, as it aligns with previous *in vitro* electrophysiological data showing that Tke5 senses the directionality of the membrane potential and is expected to preferentially insert into the inner membrane from the periplasmic side (Velázquez *et al*, 2025).

To pinpoint the toxic component of Tke5, we have systematically analysed the series of deletion constructs generated. Our findings show that the C-terminal β -rich region is not essential for toxicity, as its deletion (PelB- $\alpha^{239-863}$ or MIX- α^{1-863}) does not eliminate the toxic effect (Fig. 5c-d). Furthermore, constructs lacking the N-terminal MIX domain (PelB- α - $\beta^{239-996}$ or PelB- $\alpha^{239-863}$) remain as toxic as the full-length protein PelB-MIX- α - β^{1-996} , while the MIX domain (PelB-MIX $^{1-238}$) or the C-terminal β -rich region (PelB- $\beta^{864-996}$) alone are non-toxic (Fig. 5c). Crucially, the minimal construct containing only the α -helical region (PelB- $\alpha^{239-863}$) is sufficient to induce cell death, demonstrating that this domain is the core toxic module of the effector (Fig. 5c).

To validate that this observed toxicity is the result of Tke5's specific activity, we have co-expressed the toxic constructs with the cognate immunity protein (Fig. 5d). The presence of Tki5 fully protected the cells from the effects of full length Tke5 (MIX- α - β^{1-996}).

⁹⁹⁶, PelB·MIX- α - β ¹⁻⁹⁹⁶), and deletion constructs containing the α -helical region (MIX- α ¹⁻⁸⁶³, PelB- α ²³⁹⁻⁸⁶³). Taken together, these results confirm that the immunity protein acts by directly neutralising the α -helical region.

In summary, while our previous work identified Tke5 as a bactericidal toxin that causes cell depolarisation, our current findings establish that this activity is mediated entirely by its α -helical region. This region alone is sufficient to kill intoxicated bacteria, and the ability of Tki5 to neutralise its effect confirms that the immunity protein directly counteracts the effector's toxic activity rather than interfering with other domains."

New Figure 5:

Fig. 5 | The α -helical region of Tke5 mediates toxicity and is neutralised by its cognate immunity protein Tki5. (a) Domain architecture of Tke5 showing the N-terminal MIX domain (residues 17–238, blue), α -helical pore-forming region (residues 239–863, grey and green), and C-terminal β -rich region (residues 864–996, green). Predicted transmembrane helices (green) are mapped onto the α -helical region, which corresponds to residues 622-644 (FLGLTSGGGGLNAGMLWFNIL), 663-685 (LGFASSVFGVIGAAAATLVSV-RA), 695-717 (ISTAPGMAFGNGIINFLTSNLFA), 744-766 (TALGSILVL) and 781-803 (FAGWTAAGIALIGATLIGGGLFL). **(b)** Two 180° rotated views of the β -rich region that presents an immunoglobulin-like sandwich fold located at the C-terminal of Tke5. **(c)** Toxicity assays of PelB-containing Tke5 and truncation constructs in *P. putida*. Full-length Tke5 (PelB·MIX- α - β ¹⁻⁹⁹⁶), the α + β domain (PelB- α - β ²³⁹⁻⁹⁹⁶), and the α -helical region alone (PelB- α ²³⁹⁻⁸⁶³) show comparable toxicity. The MIX domain (PelB·MIX¹⁻²³⁸) and β -rich region (PelB- β ⁸⁶⁴⁻⁹⁹⁶) are non-toxic, with bacterial growth comparable to that of bacteria transformed with the empty vector (EV). **(d)** Neutralisation of Tke5 toxicity by its cognate immunity protein Tki5. Co-expression of Tki5 rescues cell growth inhibited by PelB·MIX- α - β ¹⁻⁹⁹⁶, PelB- α ²³⁹⁻⁸⁶³, MIX- α - β ¹⁻⁹⁹⁶ and MIX- α ¹⁻⁸⁶³, confirming the specificity of Tki5 for the α -helical toxic domain.

Expression of constructs $MIX-\alpha-\beta^{1-996}$ or $MIX-\alpha^{1-863}$ that lack the PelB leader sequence is also toxic, although to a lesser extent than PelB-containing constructs. A schematic representation of the corresponding constructs is shown next to the corresponding plot. Plotted data are the average of 3 or 6 independent biological replicates ($n = 3, 6$). Error bars correspond to standard deviations (SD). Source data for this figure are available online.

4. Related to comment 3 above, are Tke5 residues 600-800 sufficient for toxicity if fused to the PelB sequence?

Response removed as the results presented here are unpublished.

Figure for referee with unpublished data and its description has been removed upon request by the authors.

5. I'm not sure that the final section "Proposed Tke5's pore-forming molecular mechanism" is appropriate for the Results. This section is purely speculative and based solely on AlphaFold modeling. As an aside, the model confidence is low (pLLDT <70) to very low (pLLDT <50) for the putative TMD in Fig. 6b. Maybe these passages could be incorporated into the Discussion.

Thanks for the suggestion. We have moved this section to the discussion. Also, we have included a supplementary figure (**Appendix Figure S13**) showing the AF3 predicted model confidence of Tke5⁶⁰⁸⁻⁹⁹⁶. The AF3 model is colour-coded based on the predicted Local Distance Difference Test (pLDDT) score. Also, we show the Predicted Aligned Error (PAE) plot for Tke5⁶⁰⁸⁻⁹⁹⁶ that shows the expected positional error in angstroms for each pair of residues. As indicated by the pLDDT score and the corresponding PAE plot, the model confidence goes from very high (pLDDT > 90) to high (90 > pLDDT > 70) with PAE values between 0-5Å for a large portion of the predicted transmembrane helices (region containing residues 630 to 870) and the immunoglobulin like fold (Ig-like fold; region containing residues 911 to 996). The N-terminal residues that are

predicted to form a strand and a short helix (residues 608 to 629) and the loop that protrudes away from the Ig-like fold have very low pLDDT scores (pLDDT < 50). These low values might indicate highly flexible regions. Experimental determination of the Tke5 structure in its membrane bound conformation is required to validate this prediction. Please see below for the new supplementary figure. We prefer to discuss the AF3-predicted model confidence in the **Appendix Figure S13** caption to make the text more accessible to non-specialist readers.

Appendix Figure S13. AF3 model confidence for the predicted pore state of Tke5⁶⁰⁸⁻⁹⁹⁶. (a) The AF3-predicted structure of Tke5⁶⁰⁸⁻⁹⁹⁶ is inserted into a gram-negative inner membrane model, as suggested by PPM3.0 based on a membrane orientation prediction. The structure is color-coded according to the predicted Local Distance Difference Test (pLDDT) score, a metric indicating the confidence in the predicted atomic positions. The color scale is as follows: blue indicates very high confidence (pLDDT > 90); light blue, high confidence (90 ≥ pLDDT > 70); yellow, moderate confidence (70 ≥ pLDDT > 50); and orange, low confidence (pLDDT ≤ 50). (b) Predicted Aligned Error (PAE) plot for Tke5⁶⁰⁸⁻⁹⁹⁶ showing the expected positional error in angstroms for each pair of residues. Dark blue indicates very low predicted error (< 5 Å) and high confidence; yellow indicates moderate confidence with predicted error < 10 Å; orange represents low confidence with predicted error < 15 Å; gray indicates higher error and lower confidence; and white corresponds to very high predicted error (> 30 Å). The reliability of this *in silico* model as evaluated using the per-residue confidence score (pLDDT) and the Predicted Aligned Error (PAE) plot reveals that a significant portion of the model is predicted with high to very high confidence. Specifically, the regions corresponding to the putative transmembrane helices (residues 630 to 870) and the C-terminal immunoglobulin-like (Ig-like) fold (residues 911 to 996) exhibit pLDDT scores largely between 70 and 90, with many exceeding 90. The corresponding PAE plot for these domains shows low expected positional error, with values between 0-5Å, indicating high confidence in the predicted packing and relative orientation of these core structural elements. In contrast, the N-terminal residues (608 to 629), predicted to form a strand and a short helix, along with a loop region protruding from the Ig-like fold, are assigned very low confidence scores (pLDDT < 50). Such low pLDDT values are often indicative of intrinsically disordered or highly flexible regions that do not adopt a single, stable conformation.

Furthermore, we have extended the Discussion section to discuss the structure of Ptx2, which became available during the revision of this manuscript. Also, during the revision of the manuscript, a bioRxiv paper for another family member (Bte2) became available, so we have also included the relevant findings in our discussion. Please see below for details:

Lines 396-435: "The recent cryo-EM structure of *P. aeruginosa* Ptx2 revealed a "Flexible Region" (FR) within its α -helical domain, proposed to function as a membrane sensor that initiates a conformational change upon contact with the lipid bilayer, thereby exposing its transmembrane domain (TMD). Critically, this FR was not resolved in the Ptx2 cryo-EM density, a fact attributed to a high degree of intrinsic flexibility consistent with its proposed role as a dynamic sensor.

Notably, the corresponding region in Tke5 (residues 319-365) is well-resolved in our cryo-EM structure. In our model of Tke5's pre-pore state (Fig. 6a), the "FR" region is positioned close to the membrane surface. This suggests that, analogous to what has been proposed for Ptx2 (Colautti *et al*, 2025), this region of Tke5 might also be involved in the large-scale conformational rearrangement of the α -helical region that leads to the insertion of the TMD and subsequent pore formation.

Further illustrating the family's adaptability, the recent characterisation of the divergent homolog Bte2 from *Bacteroides fragilis* revealed a highly specialised, receptor-dependent mechanism (Ratner *et al*, 2025). Ratner *et al*. demonstrated that Bte2 intoxication is strictly dependent on the presence of the Rnf electron transport

complex in the target cell's inner membrane. Crucially, the catalytic activity of the Rnf complex was shown to be dispensable for toxicity, confirming that Bte2 co-opts the complex as a physical receptor to facilitate its membrane insertion. Bte2 also contains a C-terminal immunoglobulin-like fold (Appendix Fig. S11), but currently, no experimental data show which region of Bte2 is involved in binding to the Rnf receptor.

Together, these findings suggest that the BTH_I2691 family employs a modular, evolutionarily adaptable mechanism for membrane insertion, likely involving two key regions: an internal "Flexible Region" (FR), as described for Ptx2, and a C-terminal Receptor-Binding Domain (RBD), as observed in Tke5. We propose a unified model in which the FR-like region serves as the primary membrane-sensing and insertion-triggering module, consistent with findings that this region is essential for Ptx2's toxicity.

The C-terminal RBD likely functions as an accessory module that provides an enhanced tethering mechanism. In toxins like Tke5, the RBD can mediate an initial, high-avidity interaction with the membrane, thereby increasing the toxin's local concentration and positioning the FR-like region to efficiently trigger the conformational changes required for pore formation. This two-tiered model provides a framework that integrates available functional data. The fact that the Tke5 RBD is dispensable for toxicity when expressed in the cytoplasm of sensitive bacteria suggests that the FR-like region can independently initiate membrane insertion. In contrast, the RBD's demonstrated ability to bind membranes *in vitro* supports its role in enhancing this process. This modularity allows for a spectrum of targeting strategies within the family, from generalist membrane-sensing driven primarily by the FR to more specialised, RBD-assisted targeting."

Dear Dr. Albesa-Jové,

Thank you again for the submission of your revised manuscript (EMBOJ-2025-121682R) to The EMBO Journal for our consideration, and for your patience during peer review. Your manuscript has been sent back to the four original referees who had previously assessed the first version of the work, and we have now received their comments, which are appended below.

I am very pleased to say that all four referees are satisfied with the revision, mention that the original referee comments have been extensively and satisfactorily addressed, and recommend publication without further comments. In light of this input, I am glad to inform you that your manuscript has been accepted in principle for publication in our journal - congratulations on an excellent work!

From the editorial side, there are a few changes we need you to make in a final version of your manuscript, before we can move forward with its formal acceptance and publication in The EMBO Journal:

- Please note that the funding information provided in the Acknowledgements section of your manuscript should match that entered in our online manuscript handling system; currently, the following information is missing from the online system: "Project IU16-014045 (CRYO-TEM) by the Generalitat de Catalunya and by "ERDF A way of making Europe", a European Union initiative; the Spanish Ministry of Science and Innovation (MCIN/AEI/10.13039/501100011033/ FEDER, UE projects CNS2022-135585)".
- You can provide two more keywords after the Abstract if you wish (the limit is 5 keywords; preferably, these should be broad terms that would enhance the online search discoverability of your article).
- Please place your "Disclosure and competing interests statement" above the References list.
- Callouts for the missing Figures and Tables "Supplementary Fig. 1-2", "Supplementary Fig. 10a-b" and "Supplementary Table 1" should be corrected as appropriate.
- We also noticed that callouts for Fig. 4A and Appendix Figure S13 are missing.
- Regarding the Appendix PDF file: its title page should contain heading "Appendix for:", followed by the manuscript's title and a Table of Contents including page numbers for all listed items; the word "SUPPLEMENTAL" should not be used for Appendix Figures, please remove it from the Table of Contents; the Appendix references list must be reformatted (that is, it should be ordered by author name rather than being numbered; the first 10 co-authors of each reference should be listed, followed by "et al.").
- Thank you for providing the source data for your manuscript along with a completed checklist. We kindly request that the source data files be saved and uploaded in a single ZIP folder per main Figure (e.g., all source data files for the panels of Figure 1 need to be saved in a single folder, which must be zipped and uploaded as "SD Figure 1.zip" file). For EV and/or Appendix Figures, please ZIP together all source data.
- Please make sure that all deposited datasets in external repositories will be publicly available at the time of publication. The specific and permanent URLs to the deposited datasets (PDB code id: 9R8G, EMDB accession code id: EMD-53820, and the Electron Microscopy Public Image Archive – EMPIAR accession code id: EMPIAR-13034) must be included in the Data availability statement of the revised manuscript.
- Please note that EMBO press papers are accompanied online by:
 - A) a short (2 sentences) summary of the findings and their significance,
 - B) 2-5 short bullet points highlighting the key results, and
 - C) a synopsis image in .jpg or .png format that is exactly 550 pixels wide and 300-600 pixels high (the height is variable). Please note that all text needs to be legible at the final size.Please upload this information along with your revised manuscript (the text for A and B should be provided in a separate Word file).
- Please note that the definition of the scale bar is missing from the legend of Figure 6b.
- The uploaded Movie file should be renamed to "Movie EV1" and its corresponding callout updated accordingly; its legend should be zipped together with the movie file.
- The manuscript sections need to be named and ordered as follows: Title page - Abstract - Keywords - Introduction - Results - Discussion - Methods - Data Availability - Acknowledgements - Disclosure and Competing Interests Statement - References - Figure Legends - main Tables (if there are any) - Expanded View Figure Legends.

Please also note that as part of the EMBO Press transparent editorial process, The EMBO Journal publishes online a Peer Review File along with each accepted manuscript. This File will be published in conjunction with your paper and will include the referee reports, your point-by-point responses and all pertinent correspondence relating to the manuscript. Please note that your Author's Checklist will also be published at the end of the Peer Review File. Please let us know in case you want to remove any data or figures from your point-by-point responses before they are published as part of the Peer Review File. Retaining unpublished data in the Peer Review File means that these count as published and that the Peer Review File would need to be referenced in future publications. Please let the editorial office know in case you want to remove any data from this file (contact@embojournal.org).

We look forward to seeing a final version of your manuscript as soon as possible. Please let us know if you have any questions and use this link to submit your revision: Link Unavailable

Best regards,

Ioannis

Referee #1:

The authors have extensively addressed the comments from all four reviewers, and consequently the manuscript is greatly improved. I am delighted to recommend its publication in EMBO.

Referee #2:

I thank the authors for their additional work. My comments were addressed to my satisfaction.

Referee #3:

The authors have satisfactorily addressed all of my comments.

Referee #4:

The authors have addressed all of my concerns adequately.

All minor editorial requests have been addressed by the authors.

Dear Dr. Albesa-Jové,

Congratulations on an excellent manuscript! I am very pleased to inform you that it has been accepted for publication in The EMBO Journal. Thank you for comprehensively addressing the initially raised concerns of the referees and all editorial requests for changes and corrections.

You may qualify for financial assistance for your publication charges - either via a Springer Nature fully open access agreement or an EMBO initiative. Check your eligibility: <https://link.springer.com/journal/44318/how-to-publish-with-us>

If you have any questions, please do not hesitate to contact the Editorial Office. Thank you for your contribution to The EMBO Journal. Working with you has been a pleasure.

Best regards,

Ioannis
